# Blockchain technology and application: an overview

Shi Dong[1], Khushnood Abbas[1], Meixi Li[1] and Joarder Kamruzzaman[2]

[1] School of Computer Science and Technology, Zhoukou Normal University, Zhoukou, Henan, China
[2] School of Science, Engineering and Information Technology, Federation University Australia, Ballarat, Australia

## ABSTRACT

In recent years, with the rise of digital currency, its underlying technology, blockchain, has become increasingly well-known. This technology has several key characteristics, including decentralization, time-stamped data, consensus mechanism, traceability, programmability, security, and credibility, and block data is essentially tamper-proof. Due to these characteristics, blockchain can address the shortcomings of traditional financial institutions. As a result, this emerging technology has garnered significant attention from financial intermediaries, technology-based companies, and government agencies. This article offers an overview of the fundamentals of blockchain technology and its various applications. The introduction defines blockchain and explains its fundamental working principles, emphasizing features such as decentralization, immutability, and transparency. The article then traces the evolution of blockchain, from its inception in cryptocurrency to its development as a versatile tool with diverse potential applications. The main body of the article explores fundamentals of block chain systems, its limitations, various applications, applicability *etc*. Finally, the study concludes by discussing the present state of blockchain technology and its future potential, as well as the challenges that must be surmounted to unlock its full potential.

## INTRODUCTION

Blockchain is a tamper-proof distributed ledger technology. Its distributed yet secure nature makes it widely adoptable, similar to Internet technology. The fundamental operation of blockchain technology is that it enables a group of people to record transactional records in a shared ledger in such a way that once written and published, no one can change them. Blockchain technology has emerged as a transformative innovation with the potential to revolutionize various industries (*Sapra & Dhaliwal, 2022*; *Bodkhe et al., 2020*; *Mohamed & Al-Jaroodi, 2019*; *Chen et al., 2022*; *da Silva et al., 2020*; *Guo, Sun & Lam, 2023*; *Siedlecka-Lamch & Szymoniak, 2023*; *Kooshari & Fartash, 2023*; *Zaghdoud et al., 2023*; *Gad et al., 2022*). For example, it has found applications in the fashion industry (*Guo, Sun & Lam, 2023*), Flying Ad-Hoc Networks (*Zaghdoud et al., 2023*), medical field (*Siedlecka-Lamch & Szymoniak, 2023*; *Ramzan et al., 2023*), software engineering (*Kooshari & Fartash, 2023*), cloud/fog computing (*Mullick, Großmann & Krieger, 2023*;

Corresponding author
Shi Dong, njbsok@163.com



*Habib et al., 2022*), museum protection (*Bilogrivic & Stublic, 2023*), electronic invoicing (*Zhang & Lu, 2022*), human resource management (*Balon, Kalinowski & Paprocka, 2022*), international trade (*Xing, Peng & Liang, 2022*), distributed robot control (*Kumar et al., 2022*), and more. Its foundational feature, a shared and immutable ledger, has unlocked new opportunities for secure and transparent transactions involving both tangible and intangible assets. Businesses leverage blockchain to streamline operations, track valuable assets, and enhance trust and efficiency in their transactions. This technology's ability to provide quick and unalterable information in real-time fosters transparency and trust, enabling businesses to identify innovative ways to enhance their operations.

## Background and motivation

Blockchain technology gained significant public attention with the introduction of cryptocurrency, Bitcoin, in 2006. In cryptocurrencies like Bitcoin, the transfer of digital assets, often referred to as electronic cash (e-cash), occurs in a distributed manner. Bitcoin has undergone rapid evolution, drawing upon various disciplines including mathematics, cryptography, and computer science. Central to its operation is decentralization, which enables peer-to-peer transactions, coordination, and collaboration facilitated by features such as timestamps, distributed consensus, data encryption, and economic incentives within the distributed system (*Zhu, Guo & Zhang, 2021*). These features offer a compelling solution to the inefficiencies and security concerns associated with centralized data storage, making blockchain technology particularly attractive to financial intermediaries and government agencies (*Al-Jaroodi & Mohamed, 2019*; *Bodkhe et al., 2020*; *Javaid et al., 2021*). Bitcoin's success has also paved the way for the development of numerous other cryptocurrencies, such as Ethereum, signifying new possible directions for currency exchange. The blockchain technology solution for cryptocurrency opening opportunity for other fields also which we will explore later.

## Problem statement

Blockchain technology is currently gaining momentum across various industries, holding the promise of modernizing our economic system. However, it also faces several significant challenges, including scalability, energy consumption, interoperability, and regulatory concerns. Unfortunately, only a limited amount of work has been undertaken in this direction thus far. In our research, we have extensively explored nearly every aspect of blockchain, ranging from its fundamental construction to various applications. Our focus has been on examining the fundamental building blocks of blockchain technology and shedding light on various security aspects related to blockchain-based systems. The objective of this work is to provide a comprehensive review of blockchain technologies, their applications, security and privacy issues, and the research obstacles that lie ahead (*Pieters, Kokkinou & van Kollenburg, 2022*).

The remainder of this article is structured as follows: "Survey Methodology" discusses the survey methodology. "Literature Review" presents the literature review. "Key Technologies for Blockchain" covers the key technologies required to build a blockchain system. "Types of Blockchains" presents the type of blockchain. In "Security Measures in

Blockchain Systems: Un-Derstanding Attack Dynamics", we shed light on the advantages and disadvantages of blockchain systems. In "Introduction to Future Research Methods and Application Areas", we examine possible future methods and applications of blockchain systems. In "Limitations of Block Chain Technology", we discuss the limitations of blockchain systems. Finally, in "Conclusions", we conclude the article.

## SURVEY METHODOLOGY

This study follows the established rules for conducting systematic literature reviews very carefully. Our research approach involves three essential phases that are meticulously planned to ensure the completeness and rigor of our investigation: planning, data collection, and data review.

In the first phase of data collection, we conducted a thorough search using Google Scholar and the Web of Science to find academic papers related to our study on blockchain technology. This initial search aimed to include a wide range of literature on the topic. After this broad search, we took a more focused approach. Specifically, we looked closely at articles that had received a significant number of citations and directly related to the main themes of our research. Because the blockchain field is rapidly changing, we went a step further to ensure the comprehensiveness of our investigation. We expanded our search beyond academic papers to include insights from relevant websites, authoritative blogs, and technical reports. These additional sources helped us gain a comprehensive understanding of the subject matter, incorporating practical experiences and expert perspectives from those actively involved in blockchain technology from industries also. Our rigorous methodological approach highlights our dedication to conducting a thorough, rigorous, and comprehensive review. By incorporating both academic research and real-world practical insights, we aim to provide a detailed and well-rounded portrayal of the diverse landscape of blockchain technology, recognizing its dynamic evolution and its relevance across various domains.

## LITERATURE REVIEW

In early 2016, the Central Bank of China stated its intention to actively promote the official publication of digital currency. As a result, more and more financial research institutions started to take notice of blockchain technology, the innovative technology behind digital currencies (*Pilkington, 2016*). Around the same time, the UK government released a special report on blockchain technology titled "Distributed Accounting Technology: Beyond Blockchain" in an effort to vigorously develop the use of blockchain in the government sector (*Hancock & Vaizey, 2016*). *Mckinsey Company (2016)* has reported that blockchain technology is the core technology most likely to trigger a disruptive revolution, which will be the fifth wave of disruptive revolution after steam engine, electricity, information, and Internet technology (*Hancock & Vaizey, 2016*). In Asia, some Internet giants have also started researching blockchain technology and its potential applications. For example, Baidu Finance and Huaneng Trust, and Changan New Life received recognition for the country's first domestically-backed blockchain-based project. Jingdong Group has built the Jingdong Anti-Counterfeit Traceability Platform using blockchain technology. Biggest

internet corporations such as Tencent and Amazon, have developed the Tencent Blockchain Supply Chain and Amazon Managed Blockchain (AMB), respectively, funding solutions to help small businesses, microbusinesses, home-based businesses, and solopreneurs with their funding issues. Alibaba Group has leveraged the decentralized, tamper-proof, and distributed storage features of blockchain technology to launch several applications such as "AntChain" and "Trust Made Simple" (*Kong, 2021*). In some of the game's modules, which were released as part of Tencent's first Augmented Reality (AR) exploration series of portable gaming consoles in April 2019, and Microsoft Xbox gaming console used Microsoft Azure Blockchain Service in 2018.

## Key technologies for blockchain

### Role of cryptography in blockchain technology

Cryptography plays a fundamental role in ensuring the security and integrity of blockchain technology. It is extensively employed in various aspects of blockchain to guarantee confidentiality, authenticity, and immutability of data and transactions. When a user initiates a transaction on the blockchain, cryptography is used to encrypt the transaction data, allowing only the intended recipient with the matching private key to decrypt and access the information, thus ensuring confidentiality and security. Digital signatures, created using cryptographic keys (private and public), are essential in blockchain transactions, verifying the authenticity of transactions and enabling traceability to the original sender. Moreover, cryptography is instrumental in consensus mechanisms like Proof of Work (PoW) and Proof of Stake (PoS), which secure the network against attacks and maintain blockchain integrity through cryptographic puzzles and algorithms. The use of cryptographic hash functions in blockchain generates unique, fixed-length representations of data (blocks), ensuring the linkage of blocks in the chain and facilitating detection of tampering or unauthorized alterations.

Secure key management is ensured by cryptography to protect users' private keys, granting access to digital assets and transactions, and preventing unauthorized access. Additionally, cryptographic encryption safeguards sensitive data stored on the blockchain, such as personal information and business records, ensuring confidentiality even if the data is publicly accessible. Overall, cryptography serves as the backbone of blockchain technology, providing essential tools for creating a secure, transparent, and tamper-proof decentralized system. By instilling trust and confidence among participants, blockchain technology becomes a reliable solution for various applications, including digital currencies, supply chain management, identity verification, and many others (*Bhushan et al., 2021a*; *Gupta, Gupta & Chandavarkar, 2021*; *Sabry, Kaittan & Majeed, 2019*).

### Packaging into data blocks

In the blockchain system, data is organized into blocks using a particular hashing algorithm and data structure, such as the merkle tree or binary hash tree. Each distributed node in the network takes the transaction data it receives, encodes it, and packages it into blocks of data. These blocks are then given a timestamp and linked to the longest main blockchain. This process involves various technical components, including blocks, chain

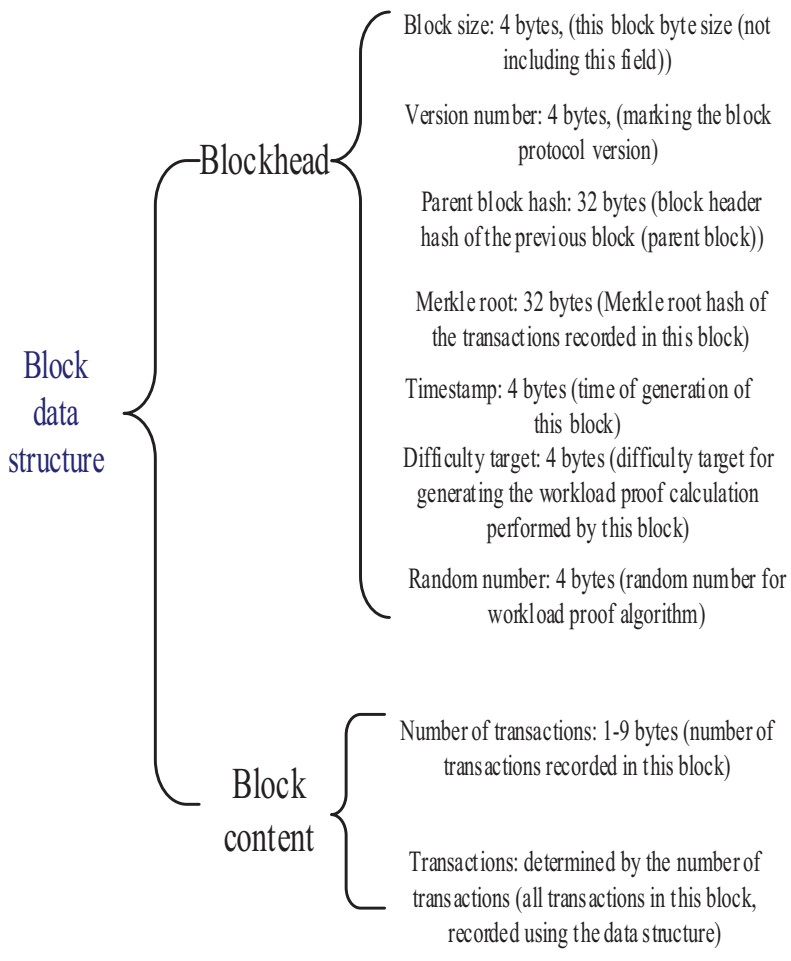

Block data structure

Blockhead
- Block size: 4 bytes, (this block byte size (not including this field))
- Version number: 4 bytes, (marking the block protocol version)
- Parent block hash: 32 bytes (block header hash of the previous block (parent block))
- Merkle root: 32 bytes (Merkle root hash of the transactions recorded in this block)
- Timestamp: 4 bytes (time of generation of this block)
- Difficulty target: 4 bytes (difficulty target for generating the workload proof calculation performed by this block)
- Random number: 4 bytes (random number for workload proof algorithm)

Block content
- Number of transactions: 1-9 bytes (number of transactions recorded in this block)
- Transactions: determined by the number of transactions (all transactions in this block, recorded using the data structure)

**Figure 1 Block data structure.**

structures, hashing algorithms, Merkle trees, and timestamps (*Zhu, Guo & Zhang, 2021*). These elements work together to ensure the secure and orderly arrangement of data within the blockchain system.

### Block

The unit of data that can record information about Bitcoin transactions is the block. A block is made up of two parts: one part is the block header, and the other part is the block content. This is shown in Fig. 1.

### Merkle tree

The Merkle tree plays a crucial role in blockchain technology, serving as a vital data structure for efficiently summarizing and verifying the existence and integrity of block data (*Mohan, Mohamed Asfak & Gladston, 2020*; *de Ocáriz Borde, 2022*). Its main function is to enable the identification of all transactions recorded in a block, making it possible to locate them on each block of the blockchain. To achieve this, the blockchain system utilizes a binary tree variant of the Merkle tree. This variant is responsible for summarizing and

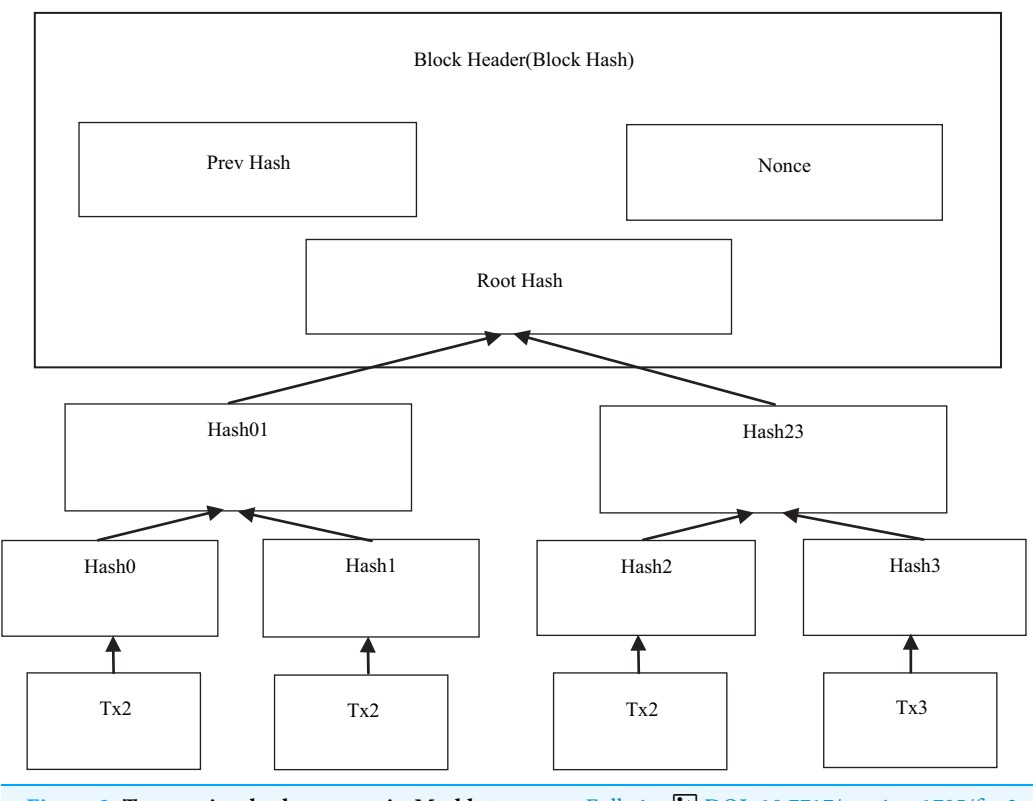

**Figure 2** **Transaction hash concept in Merkle tree.**

representing the transactions in a block, ultimately producing a digital signature for the entire set of transactions. Figure 2 provides a visual illustration of this concept.

### Timestamp

In blockchain technology, nodes with bookkeeping privileges are required to include a timestamp in the header of the current data block. This timestamp indicates the exact time when the block was written or added to the blockchain. By incorporating this timestamping mechanism, the blockchain ensures that blocks on the main chain are arranged in a chronological order, reflecting the sequential order of transactions.

The inclusion of timestamps enhances the tamper-evident nature of the blockchain. Any attempt to modify or alter the data in a block would result in a mismatch between the timestamp and the actual time of the tampering, immediately indicating the presence of unauthorized changes. This chronological organization and tamper-evident feature contribute to the overall security and integrity of the blockchain system.

### P2P network technology

Since its creation in 2009, the Bitcoin system has shown impressive operational stability, primarily attributed to its use of peer-to-peer (P2P) network technology. Unlike traditional client-server models, P2P network architectures provide several advantages, including enhanced privacy protection, decentralization, robustness, load balancing, and improved performance (*Rajasekaran, Azees & Al-Turjman, 2022*). The essence of P2P technology is rooted in decentralization. Within the context of blockchain, this architecture facilitates

the global transfer of cryptocurrencies without intermediaries or central servers. Leveraging a distributed network structure, individuals interested in validating blocks can establish a Bitcoin node (*Sharma, 2022*). Blockchain serves as a decentralized ledger that tracks digital assets on a P2P network. This network arrangement involves interconnected computers, each housing a complete ledger copy. The devices cross-reference their copies to ensure data accuracy. This diverges from traditional banking models, wherein transactions are privately stored and exclusively managed by the bank. Based on different design concepts, emergence times, and network structures, P2P networks can be classified into three distinct types: first-generation hybrid P2P networks, second-generation unstructured P2P networks, and third-generation structured P2P networks.

### Distributed ledger technology

There is a significant distinction between blockchain technology and traditional databases, mainly in terms of the fundamental operations they support (*Chowdhury et al., 2018*; *Koens & Poll, 2018*; *Mattila, 2016*). Traditional databases provide four core operations: adding, deleting, modifying, and querying data. In contrast, blockchain technology offers only two operations: adding and querying data. Notably, blockchain lacks the capability to modify or delete data once it is recorded.

Traditional databases can be categorized into two types: distributed databases and centralized databases. In distributed databases, high-speed networks connect multiple geographically dispersed data storage units, creating a logically unified database. This approach allows for the storage of large amounts of data and facilitates higher concurrent traffic.

On the other hand, blockchain technology falls under the category of distributed ledger technology. While it shares some similarities with distributed databases, there are significant differences in terms of storage mechanisms and data structures. The immutability of data in blockchain, along with its decentralized nature, ensures a secure and transparent ledger of transactions.

### Asymmetric encryption and digital signature

Asymmetric encryption relies on a matched pair of public and private keys, which exhibit unique characteristics. Public-private key pairs are created systematically, wherein a key pair consists of a public key and its corresponding private key. A fundamental stipulation of this process is that, while the public key is openly accessible to all, the private key remains non-derivable even when the public key is known. The encrypted with the public key can only be decrypted using the corresponding private key (*Bhushan et al., 2021b*). Similarly, content encrypted with the private key can only be decrypted using the corresponding public key. To ensure secure message transmission, the digital signature method is utilized (*Shi et al., 2020*). The message sender applies a hash operation to the message digest and subsequently attaches the resulting digital signature at the end of the message. The message is then encrypted using the private key, and upon receiving the message, the recipient decrypts it using the public key as shown in Fig. 3. The recipient subsequently performs the same hash operation on the digest to verify that the message has

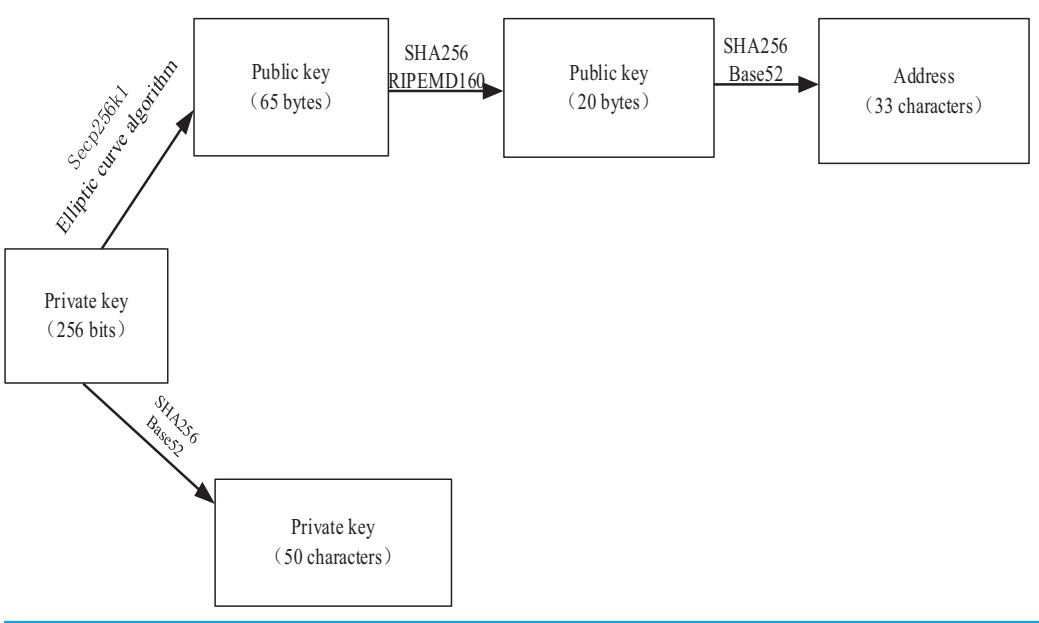

**Figure 3 Bitcoin asymmetric encryption mechanism.**

not been tampered with, as evidenced by the consistency of the digital signature (*Li, Hu & Lan, 2020*).

### Elliptic curve type encryption algorithm

Today, most blockchain researchers opt to utilize the elliptic curve cryptographic algorithm. This particular encryption method leverages an algorithm from the field of elliptic curve mathematics: (i) Despite its name, the elliptic curve cryptographic algorithm does not actually involve an elliptic curve. Rather, its expression bears a resemblance to the integral formula used to determine the circumference of an ellipse. (ii) elliptic curve formula uses the following expression-

$$q = kp \tag{1}$$

where, k (a positive integer less than p), and p is a fixed point on the elliptic curve and serves as the starting point for the scalar multiplication, and it is possible to quickly determine the value of q. However, if q and p are given, it becomes difficult to compute the value of k. This mathematical challenge is referred to as the discrete logarithm problem of elliptic curves. By using q as the public key, it can be safely shared with others for encryption purposes (*Umucu, 2022*). While keeping k as the private key ensures that only the authorized person can decrypt the message. The discrete logarithm problem of elliptic curves makes it challenging for an unauthorized party to obtain the private key k from the public key q, ensuring a high level of security.

### Paxos algorithm

The Paxos algorithm is a widely used algorithm that enables achieving consensus in blockchain technology (*Mingxiao et al., 2017*; *Charapko, Ailijiang & Demirbas, 2018*; *Deng et al., 2022*; *Burchert & Wattenhofer, 2018*), specifically in the presence of node failures

rather than malicious nodes (Byzantine failures). It is designed to solve non-Byzantine problems and ensure agreement among nodes even in the face of failures. The Paxos algorithm operates similarly to a parliamentary system, where proposals are presented and voted on by all members. It consists of two main phases: the preparation phase and the submission phase. In the preparation phase, voting on the proposal takes place, and in the submission phase, the final acceptance of the proposal is determined. During the commit phase, if the proposal receives a majority of affirmative responses from the nodes, the system sends an acknowledgment message, comparable to a webpage prompting you to "confirm" before closing. If all nodes confirm the proposal, it will be accepted and considered agreed upon. However, if the proposal fails to gather sufficient confirmations, a new proposal must be submitted to replace it, and the process continues until a consensus is reached among the nodes. This way, the Paxos algorithm ensures fault-tolerant consensus in distributed systems with node failures.

### SHA256 algorithm

Bitcoin uses a double SHA256 hash function to obtain a 256-bit hash from the original transaction record of any length. This hash function is advantageous due to its fixed length, timing, single direction, and randomness. Fixed length refers to the output hash values having the same length, while timing means that the time needed to compute the hash is virtually the same for different lengths of data. Single direction means that the original input data cannot be derived from the hash, although theoretically possible, it is practically impossible. Randomness means that even with similar values entered, the output hash will be completely different. Moreover, the proof of work used in Bitcoin is also based on the SHA256 function (*Ye et al., 2018*).

### Consensus mechanism

A consensus algorithm serves as a critical procedure within a blockchain network, enabling each peer to establish a unified agreement on the distributed ledger's state. Essentially, it acts as a protocol facilitating all nodes in the blockchain network to collectively determine the current data state within the ledger and trust unknown peers in the network. The blockchain network implements an incentive-based block creation process also known as "block mining" (*Wang et al., 2019b*).

The consensus mechanism stands as a foundational technology in the blockchain realm. It identifies the nodes responsible for maintaining the ledger and ensures the confirmation and synchronization of transaction information. This consensus process typically involves two key phases: "master selection" and "bookkeeping," with each round being further subdivided into four stages: master selection, block generation, data verification, and uploading (*i.e.*, bookkeeping) (*Castro & Liskov, 1999*). Presently, mainstream consensus mechanisms encompass proof of work, practical Byzantine fault tolerance (dBFT), tangle (IOTA), proof of stake (PoS), delegated proof of stake (DPoS), Ripple consensus protocol, proof of weight, proof of elapsed time, proof of history, proof of stake velocity, proof of importance, proof of reputation, proof of identity, proof of activity, proof of time, proof of retrievability, proof of capacity, Byzantine fault tolerance (BFT), delayed proof of work,

RAFT, stellar consensus, proof of believability, directed acyclic graphs, Hashgraph, proof of work (PoW), holochain, proof of existence, SPECTRE, proof of authority, ByteBall, LibraBFT (*Amsden et al., 2020*), and more come into play (*Lashkari & Musilek, 2021*; *Zhang, Wu & Wang, 2020b*; *Chepurnoy et al., 2017*; *Sanka et al., 2021*; *Guru et al., 2023*; *Pilkington, 2016*; *Kaur et al., 2021*; *Underwood, 2016*; *Mukhopadhyay et al., 2016*; *Wang et al., 2019b*; *Yao et al., 2021*). However, it is worth noting that the ever-evolving landscape of blockchain technology necessitates vigilance against potential vulnerabilities. Hackers continuously upgrade their computational capabilities, posing a future security risk to blockchain systems. These diverse consensus mechanisms showcase the innovation and adaptability within the blockchain ecosystem, addressing specific use cases and requirements. Here we are describing some well-known protocols.

- **Proof of Stake (PoS):** PoS is an alternative consensus mechanism where validators are chosen based on their stake in the network (cryptocurrency holdings). It is more energy-efficient than PoW and might be more suitable for IoT devices with limited resources.
- **Delegated Proof of Stake (DPoS):** DPoS is a variation of PoS where participants vote for a set of delegates who then validate transactions and create blocks. It offers faster transaction times and is commonly used in blockchain networks like EOS.
- **Proof of Work (PoW):** The PoW is a crucial and foundational mechanism in blockchain technology. Its primary role is to achieve consensus and secure the blockchain network by adding new blocks to the blockchain (*Gervais et al., 2016*; *Gemeliarana & Sari, 2018*; *Shi, 2016*). The mechanism's fundamental steps include: (i) nodes monitor and temporarily store network data records, which are subsequently verified for their basic legitimacy; (ii) nodes utilize their computational power to test different random numbers; (iii) after identifying a suitable random number, nodes generate the corresponding block information by first inserting the block header information, followed by the data record information; (iv) upon receiving the instruction, the newly generated block is broadcasted to the network. Once the remaining nodes pass the verification process, the block is added to the blockchain, and a node is added to the height of the main chain, increasing its height by one. The proof of work (PoW) method aims to establish a reward mechanism to incentivize other nodes in the blockchain network to solve a SHA256 mathematical problem, which is difficult to solve but easy to verify. The mathematical problem requires that the computed random number be equal to or less than the target hash value.
- **Proof of Stake (PoS):** PoS is a consensus mechanism used in blockchain networks to achieve agreement on the state of the blockchain and validate new transactions. In this model, participants stake their digital currency to become validators. The more coins they stake, the higher their chances of being chosen to create and validate new blocks (*Saleh, 2021*; *Li et al., 2017*; *Gaži, Kiayias & Zindros, 2019*; *Shifferaw & Lemma, 2021*). To incentivize holding coins and discourage hoarding, the concept of "coin days" is employed. This means that for each coin owned, one "coin day" is generated every day. For example, if someone holds 200 coins for 15 days, their total coin days would be 3,000. When a new PoS block is discovered, the individual's coin days are reset to zero.

As a reward for participating in PoS and staking their coins, certificate holders receive interest. Every time 365 coin days are cleared, the holder is entitled to receive 0.05 coins as interest from the associated blocks. For instance, with 3,000 coin days, the interest earned would amount to 0.41 coins, indicating that the currency held would accrue interest.

The fundamental idea behind the PoS model is to replace the energy-intensive Proof of Work system with a more efficient approach. Instead of selecting the node with the highest computational power, PoS selects validators based on their equity in the blockchain network. This encourages active participation and ensures that validators have a vested interest in maintaining the network's integrity while earning incentives for their contributions.

- **Delegated Proof of Stake (DPoS):** It is a consensus mechanism used in some blockchain networks to achieve agreement on the state of the blockchain and validate new transactions. DPoS is a variation of the PoS algorithm but with a different approach to selecting validators. In a DPoS system, token holders in the blockchain network have the right to vote for a certain number of delegates or representatives who will be responsible for validating transactions and creating new blocks. These delegates are often referred to as "witnesses" or "block producers (*Saad & Radzi, 2020*)".

The key features and principles of DPoS include:

1. **Voting:** Token holders in the network can vote to elect delegates from a pool of candidates. The number of votes a token holder has is typically proportional to the number of tokens they hold. The elected delegates then take on the responsibility of validating transactions and adding blocks to the blockchain.

2. **Block production:** The elected delegates are responsible for creating new blocks. They take turns in producing blocks in a round-robin fashion or based on a predefined schedule.

3. **Consensus:** Consensus is achieved when a supermajority of elected delegates agree on the validity of a transaction and its inclusion in the blockchain.

4. **Decentralization:** Although DPoS relies on a limited number of delegates, it is still considered decentralized because token holders have the power to vote and change the delegates if they are dissatisfied with their performance.

5. **Efficiency and scalability:** DPoS is known for its high transaction throughput and faster block confirmation times compared to other consensus mechanisms like PoW.

6. DPoS relies on the assumption that elected delegates act in the best interest of the network since they have a stake in it. To prevent malicious behavior, penalties or mechanisms like vote slashing can be implemented. DPoS has been adopted by several blockchain projects, including Steem, BitShares, and EOS. It aims to strike a balance between decentralization, efficiency, and security, making it suitable for applications that require high transaction speeds and scalability. However, it also introduces some degree of centralization due to the limited number of elected delegates, which is a topic of ongoing debate within the blockchain community.

**Table 1 Comparison of blockchain consensus mechanisms.**

| Consensus | Year | Type | Mining | Scalability |
|---|---|---|---|---|
| Practical byzantine consensus algorithm | 1999 | Permissioned | Round of mining | Not scalable |
| PoW | 1999 | Permissionless | Yes | Low |
| PBFT | 1999 | Permissioned | No | Very high |
| Proof of work | 2008 | Permissionless | Computational power | Not scalable |
| Proof of stake | 2011 | Permissioned and permissionless | Node wealth and staking age | Scalable |
| Ripple | 2012 | Permissioned | No | High |
| PoS | 2012 | Permissionless | Yes | High |
| Raft | 2013 | Permissioned | No | Low |
| Proof of stake velocity | 2014 | Stake and amount (velocity) | Scalable | Low |
| DPoS | 2014 | Permissionless | Yes | Very high |
| Proof of burn | 2014 | Permissioned and permissionless | Coin burning (probabilistic lottery) | Scalable |
| Proof of activity | 2014 | Permissionless | Effectiveness of work by the miner | Scalable |
| Tendermint | 2014 | Permissioned | No | Very high |
| ELASTICO | 2016 | Permissionless | No | Low |
| Implicit consensus | 2017 | Permissioned | No | Not scalable |
| Proof of vote | 2017 | Consortium | Voting mining | Very low |
| DBFT consensus algorithm | 2018 | Permissioned | Random selection of miner | Not scalable |
| Proof of trust (PoT) | 2018 | Permission-based consortium | Probabilistic and voting mining | Scalable |
| LibraBFT | 2020 | Permissioned | Voting mining | Scalable |

- **PBFT:** The Practical Byzantine Fault-Tolerant algorithm (PBFT) (*Castro & Liskov, 1999*) is a state machine replication algorithm that models services as state machines. The algorithm addresses the low-efficiency issues of the original Byzantine Fault-Tolerant algorithm and reduces the complexity from exponential to polynomial level. We have compared well-known consensus mechanism algorithms in the Table 1.

### Blockchain wallet

A blockchain wallet is a digital wallet that securely stores and manages multiple cryptocurrencies, allowing users to exchange and transfer funds with utmost security (*Dai et al., 2018*; *Eyal, 2022*). It offers privacy and identity protection and can be accessed *via* web devices. The wallet has essential features to facilitate secure and reliable transfers and exchanges between parties.

A blockchain wallet comes with a private and public key. The public key is like an email address that can be shared with anyone to receive funds. However, the private key is confidential, like a password, and should never be shared as it is used to spend the funds. If the private key is compromised, there is a high risk of losing all cryptocurrency deposits in the account (*Suratkar, Shirole & Bhirud, 2020*). The sequence diagram is described in Fig. 4. The arrows show the flow of communication between these participants. The user initiates the transaction through the wallet, which broadcasts it to the network. The network validates the transaction, and then confirms it back to the wallet, which in turn

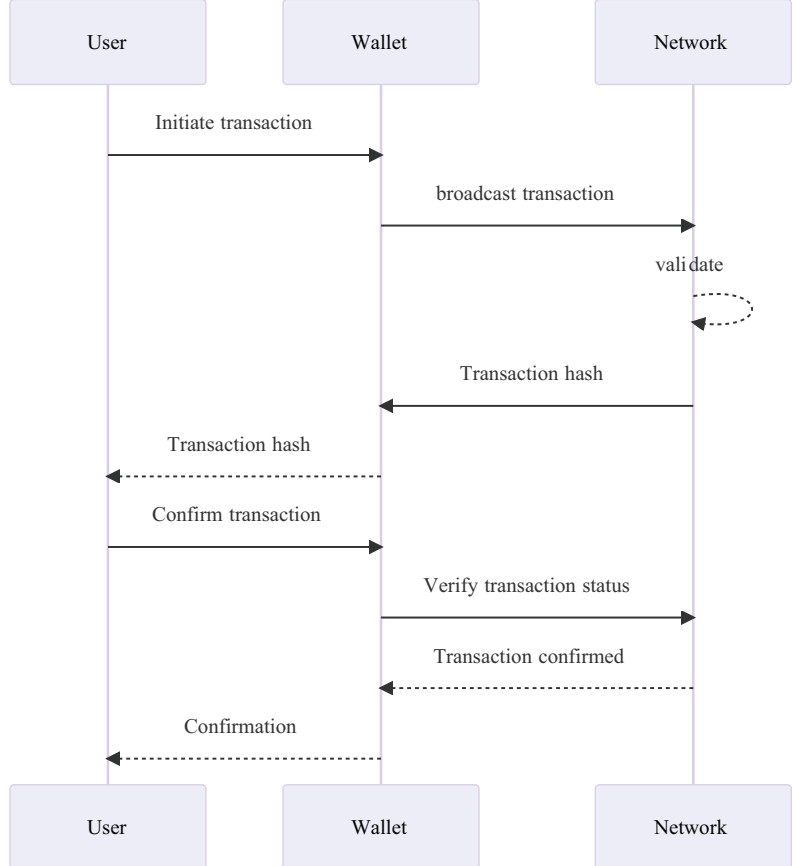

**Figure 4 The sequence diagram of blockchain technology consisting of wallets.**

confirms the transaction to the user. When a user initiates a transaction using their preferred wallet application, the transaction is broadcast to the blockchain network for processing and validation. The network will then verify the transaction using a consensus algorithm, ensuring that the transaction is legitimate and that the user has the required funds to complete it. Once confirmed, the wallet receives notification of the successful transaction, and the user can then proceed with the next step in their transactional process. Ultimately, this coordinated effort between the user, wallet, and blockchain network ensures secure and reliable transfers and exchanges of cryptocurrencies.

## Blockchain-based algorithms for IoT security

Contemporary advancements in Internet of Things (IoT) infrastructure, exemplified by entities like smart buildings and smart cities, are confronted by two significant impediments. Firstly, there exists a notable deficit of trust among the diverse constituents of the system. Secondly, a susceptibility to single point failure emerges as a vulnerability capable of inflicting severe disruptions to the entirety of the system's operation. In response to these challenges, the decentralized attributes inherent to blockchain technology present a viable solution. The distributed nature of blockchain engenders heightened system

resilience by obviating the vulnerabilities associated with single-point failures. The immutability and distributed structure of blockchain confers authenticity upon every interaction transpiring within the IoT framework. These interactions are chronologically captured as transactions, serving a dual role by both substantiating legitimate engagements and preempting dubious activities. Blockchain transactions encapsulate a spectrum of interactions, encompassing scenarios such as user movements across IoT zones, secure data transfers among devices and users, user-device engagement logs within a smart urban or architectural context, inter-organizational collaborations enhancing user service, and device-to-device interactions. This plethora of interactions is stored as a sequence of transactions, collectively crafting a coherent narrative of user-device engagement within the system. The de-centralized essence of blockchain architecture ensures that the IoT system remains resilient against compromise, thus mitigating the risks associated with single point failures. The function of blockchain within the context of the Internet of Things (IoT) entails furnishing a methodical framework for processing and preserving secure data records *via* IoT nodes. Blockchain, characterized by its robust security features, is a technology amenable to public and open utilization. Given the disparate and diverse landscape of IoT nodes, a technology of this nature is indispensable for ensuring secure communication amongst these nodes. The transactional aspects inherent to blockchain are endowed with traceability and accessibility, accessible to authenticated individuals engaging within the IoT ecosystem. Thus, the attributes inherent to blockchain render it an optimal selection for fortifying the security paradigm within IoT communication networks (*Roman, Zhou & Lopez, 2013*; *Agrawal et al., 2018*; *Alam, 2019*). Blockchain employs several methods to ensure secure communication within the realm of the Internet of Things (IoT). These methodologies contribute to fortifying the integrity, authenticity, and confidentiality of IoT communication:

### Hashing algorithms

- **Secure Hash Algorithm 256-bit (SHA-256):** This is widely used in Bitcoin and other cryptocurrencies. It provides a high level of security and is resistant to collision attacks. It ensures data integrity by generating fixed-size hash values for the data, which cannot be reversed to the original data.
- **Secure Hash Algorithm 3 (SHA-3):** The successor of SHA-256, SHA-3 is part of the Keccak family of cryptographic hash functions. It was designed to offer enhanced security and resistance against potential attacks.

### Asymmetric cryptography algorithms

- **Rivest-Shamir-Adleman (RSA):** RSA is widely used for secure key exchange and digital signatures. It provides a secure way for IoT devices to establish trust and communicate securely over a blockchain network.
- **Elliptic Curve Cryptography (ECC):** ECC offers the same level of security as RSA but with shorter key lengths, making it more suitable for resource-constrained IoT devices.

### Zero-knowledge proofs

- **zk-SNARKs (Zero-Knowledge Succinct Non-Interactive Argument of Knowledge):** zk-SNARKs allow one party to prove the knowledge of a secret without revealing the secret itself. This is useful for privacy-preserving transactions and authentication in IoT scenarios.

### Smart contract languages

- **Solidity:** Solidity is the most popular language for writing smart contracts on the Ethereum blockchain. It is widely used for creating decentralized applications (dApps) and executing business logic within the blockchain network.
- **Vyper:** Vyper is an alternative to Solidity, with a focus on simplicity and security. It is also used for writing smart contracts on Ethereum.

### Privacy-focused algorithms

- **Zero-Knowledge Scalable Transparent Arguments of Knowledge (zk-STARKs):** zk-STARKs are an evolution of zk-SNARKs, providing scalability and transparency in zero-knowledge proofs.
- **Confidential transactions:** These algorithms, like Bulletproofs, ensure transaction amounts remain confidential, enhancing privacy in blockchain networks.

### Interoperability protocols

- **Atomic swaps:** Atomic swaps allow for direct exchange of cryptocurrencies between different blockchains without the need for a trusted intermediary.
- **Polkadot:** Polkadot is a multi-chain framework that enables cross-chain communication and interoperability between different blockchains.

## Types of blockchains

The blockchains have been classified into three types based on their intended use and specific requirements: public, private, and consortium (also known as federated) blockchains. Each type of blockchain network is designed to serve a specific purpose and address particular issues, and each has its unique set of features and benefits over the others (*Guegan, 2020*).

### Public blockchain

A public blockchain is a type of blockchain that is open for all participants to read and use for transactions, and anyone can participate in the process of creating consensus. It operates without a central register or trusted third party, and the governance of public channels is based on the "Code is Law" principle that emerged from the open-source movement and cypherpunk philosophy. In this system, nodes in the network validate the choices discussed and initiated by developers by deciding whether to integrate the proposed modifications (*Karafiloski & Mishev, 2017*).

One of the significant advantages of public blockchain technology is that it is entirely trustworthy and transparent. The blockchain is a decentralized system that allows users to directly interact with each other without the involvement of intermediaries. This eliminates the need for a third party and reduces the chances of fraud or manipulation. Moreover, blockchain technology offers high-security measures due to its complex cryptographic algorithms that ensure the integrity of data stored on the blockchain. However, public blockchain technology also has its demerits. One of the most significant challenges with blockchain is scalability. As more and more data is added to the blockchain, the network becomes slower, making it difficult to process transactions quickly. Additionally, the lack of transaction speed in the blockchain system can be a drawback, especially in industries where time is of the essence. Another major concern with blockchain technology is that it consumes a lot of energy, which can be detrimental to the environment. Therefore, while blockchain technology has numerous benefits, it is important to consider its drawbacks before implementing it in various industries.

### Private blockchain

In contrast to public blockchains, a private blockchain is a more restrictive and permissioned blockchain that functions within a closed network. It is predominantly utilized within organizations where only specific members have access to the blockchain network. This type of blockchain is particularly suited for enterprises and businesses that seek to utilize blockchain solely for internal purposes. One key difference between public and private blockchains lies in their accessibility; the former is highly accessible while the latter is limited to a select group of individuals. Additionally, a private blockchain is more centralized since a single authority is responsible for maintaining the network. Notable examples of private blockchains include Corda, Hyperledger Fabric, and Hyperledger Sawtooth. Private blockchains are known for their higher transaction processing speed and scalability. They operate within a closed network of selected participants, allowing for increased processing power and efficiency. Unlike public blockchains, private blockchains are less decentralized, with a single authority maintaining the network. This centralized structure allows for higher scalability as there are no limitations to the number of nodes that can be added to the network. With greater control over the network, private blockchains can be customized to meet the specific needs of the organization, further enhancing their scalability. In addition, private blockchains offer a higher transaction per second (TPS) rate, allowing for a greater volume of transactions to be processed in a shorter amount of time. Overall, private blockchains are an ideal solution for enterprises and businesses that require high levels of scalability and transaction processing speed. Private blockchains, while offering higher transaction speeds and scalability, have some notable demerits. The first of these is that private blockchains are less secure compared to public blockchains. This is because the private blockchain operates within a closed network and is more centralized than public blockchains. As a result, it is more vulnerable to attacks by hackers and other malicious actors. Another demerit is that private blockchains are less decentralized compared to public blockchains. Achieving trust in a private blockchain can be difficult as a result. As such, private blockchains are not suitable for use cases that

require a high level of security and trust. Despite these demerits, private blockchains remain a valuable tool for businesses and organizations that require a closed blockchain network for internal use cases.

### Consortium or federated blockchain

Consortium blockchain, also known as federated blockchain, is ideal for organizations that require both public and private blockchains (*Sheth & Dattani, 2019*). In this type, multiple organizations are involved and responsible for providing access to specific nodes for reading, writing, and auditing the blockchain. As there is no single authority governing the control, it is partially decentralized. It falls between fully public (fully decentralized) and private (fully centralized) blockchain networks.

Consortium blockchain, has been gaining popularity among organizations due to its unique benefits. One of the most significant advantages is that it is best suited for collaboration between organizations. Consortium blockchain is designed to allow multiple organizations to work together on a shared blockchain network while maintaining their privacy and confidentiality. Another benefit is its scalability and high level of security, which makes it a better choice over public blockchains. Furthermore, consortium blockchains are much more efficient than public blockchains, making them ideal for use cases that require high transaction speeds. Additionally, organizations using consortium blockchains can have better customizability and control over resources, enabling them to tailor the blockchain to their specific needs. All of these advantages make consortium blockchains an excellent choice for organizations looking to collaborate and share information in a secure, efficient, and customizable manner.

While Consortium blockchains offer advantages such as scalability, security, and customizability, there are also some drawbacks to consider. One disadvantage is that consortium blockchains may be less transparent than other types of blockchains. This is because access to the blockchain network is restricted to selected participants, and the information contained in the blockchain may not be visible to everyone. Another drawback is that consortium blockchains may be less anonymous than other blockchains, as the participants in the network are pre-selected and known to each other. Therefore, consortium blockchains may not be the best option for use cases where anonymity is a critical requirement.

Table 2 presents a summary of the functioning of three distinct types: public, private, and consortium. Public blockchains are open to all and permissionless, allowing for decentralized governance and consensus mechanisms such as proof of work and proof of stake. While they offer high security, they have limited scalability and are generally slower in terms of efficiency. Private blockchains, on the other hand, are restricted to authorized users and offer better efficiency and scalability, thanks to their permissioned governance and consensus mechanisms. Consortium blockchains offer shared control among authorized participants, allowing for collaborative projects between organizations. They also offer better efficiency and scalability than public blockchains, while still maintaining high security. The use cases for each type of blockchain include cryptocurrency and

**Table 2  Classification of blockchain technology based on their applications.**

| Feature | Public blockchain | Private blockchain | Consortium blockchain |
| --- | --- | --- | --- |
| Access | Open to all | Restricted to authorized users | Restricted to authorized participants |
| Permission | Permissionless | Permissioned | Permissioned |
| Governance | Decentralized | Centralized | Shared control |
| Consensus | Proof of work, proof of stake | Various consensus mechanisms | Various consensus mechanisms |
| Scalability | Limited | Better than public | Better than public |
| Security | High | High | High |
| Efficiency | Slow | Fast | Fast |
| Use cases | Cryptocurrency, decentralized apps | Internal enterprise use | Collaborative projects between organizations |

decentralized apps for public blockchains, internal enterprise use for private blockchains, and collaborative projects between organizations for consortium blockchains.

# SECURITY MEASURES IN BLOCKCHAIN SYSTEMS: UNDERSTANDING ATTACK DYNAMICS

Blockchain security encompasses strategies, protocols, and mechanisms that protect against unauthorized access, data breaches, and malicious actions. Key focus areas include integrity, confidentiality, and network availability. Security measures counter various attacks, ensuring the trustworthiness of blockchain data. We have also summarized in Table 3.

## Security

Blockchain technology's remarkable security stems from its capability to safeguard information interactions from human intervention through decentralized operations, consensus mechanisms, and immutability. Blockchain operates on a decentralized network of nodes, ensuring no single entity has full control over the data. Consensus mechanisms require agreement among participants for data validity, preventing malicious intervention. Once data is recorded on the blockchain, it becomes tamper-proof and immutable due to cryptographic links. Participants' private keys secure their data, authorizing transactions and safeguarding against unauthorized access. Blockchain's transparency enables auditing and immediate detection of unauthorized intervention, enhancing trust and data integrity (*Sun, Zhang & Han, 2023*). Overall, blockchain's inherent features ensure secure and trustworthy information interactions. Participants in blockchain interactions play a vital role in ensuring effective information security by safeguarding their private keys (*Alangot et al., 2020*). To delve deeper into its security aspects, this section will provide a concise explanation of various attack types and the tactics employed by both honest miners and attackers within the security assessment model (*Zeng et al., 2019*). Understanding these intricacies is crucial for comprehending the robust security framework of blockchain technology.

**Table 3 Advantage and disadvantage of blockchain.**

| Advantages | Disadvantages |
| --- | --- |
| Security: enhanced data protection through cryptographic techniques and decentralization. | Scalability: potential performance issues due to distributed nature. |
| Transparency: Distributed ledger ensures all participants have access to the same information. | Energy consumption: proof-of-work consensus mechanisms can be energy-intensive. |
| Decentralization: no central authority, reducing the risk of single point of failure. | Lack of regulation: regulatory uncertainties and potential misuse. |
| Immutability: tamper-resistant data records once recorded on the blockchain. | Irreversibility of transactions: difficult to undo incorrect transactions or fraud. |
| Fast and low-cost transactions: peer-to-peer transactions without intermediaries. | Complexity and learning curve: technical expertise required for understanding. |
| Traceability and audibility: timestamped and transparent transaction history. | Data storage: expensive and impractical for large data storage. |
| Smart contracts: automated self-executing contracts with predefined conditions. | Limited throughput: longer transaction processing time in some blockchains. |
| Enhanced privacy: pseudonymity for added privacy. | Lack of governance mechanisms: slow decision-making and consensus challenges. |
| Global accessibility: internet-based accessibility for anyone with an internet connection. | Security vulnerabilities: potential vulnerabilities in implementation. |
| Reduced fraud: tamper-resistant nature reduces fraudulent activities. | Interoperability challenges: difficulty in transferring data between different blockchains. |

## Various kinds of attacks

In the realm of blockchain technology, an attack pertains to any action aimed at undermining the integrity or safety of the blockchain network. Diverse kinds of attacks are possible, including but not limited to 51% attack, Sybil attack, DDoS attack, and replay attack, among others. Such attacks may be carried out by cybercriminals or malicious parties seeking to take advantage of the blockchain network's weaknesses to access unauthorized information, steal digital assets, or cause network disruptions. To forestall attacks on the blockchain network, a range of security measures are implemented, including consensus mechanisms, cryptographic protocols, and network monitoring tools. It is imperative for blockchain developers and users to be cognizant of the potential security risks and take necessary measures to ensure the blockchain network's safety and security. Here we are describing few attacks as described by researchers (*Yaga et al., 2018*; *Malik et al., 2019*) follows:

### 51% attack

One of the most widely recognized types of attack in the blockchain ecosystem is the 51% attack. In this attack, the attacker controls more than 51% of the computing resources in the entire blockchain network, enabling them to prevent the confirmation of a new transaction and interrupt the user's transaction process. The attacker can quickly confirm false transaction information and create a longer blockchain with that information appearing more frequently. The greater the number of computing resources under the attacker's control, the easier it is to execute the attack. The 51% attack can lead to the

double payment problem by altering previously confirmed transaction information. *Nakamoto (2008)* presented the Bitcoin principle and assessed the probability of successful attacks under various attack forces. In this context, honest chains are compared to attack chains and are akin to random walk processes, as described by a binomial distribution. The probability that an honest chain can be caught up by an attacking chain is shown in Eq. (2).

$$q_z = \begin{cases} 1, p \leq q \\ (q/p)^z, p > q \end{cases} \tag{2}$$

where $p$ is the probability that the next block will be found by an honest miner, q is the probability that the next block will be found by the attacker, and q is the probability that the current block transaction can be changed by the attacker. From a probabilistic standpoint, if the honest miner cannot match the attacker's rate of computing resources, the attacker can manipulate the transaction content. In fact, the likelihood of an attack is determined by several factors, including the difficulty of the proof-of-work (PoW) algorithm, the variations in computing resources in the network, and the synchronization of transaction information between miners.

### Double-spending

In the realm of blockchain technology, a double spending attack is a grave threat that arises from the ability of a user to spend a digital asset, such as cryptocurrency, more than once. This exploit is based on the inherent replicability of a digital asset, enabling it to be simultaneously used in multiple transactions. To perform this attack, the attacker initiates a transaction and rapidly creates another transaction using the same digital asset but with a higher transaction cost. Subsequently, the attacker attempts to get the second transaction confirmed by the network faster than the first one, leading to the confirmation and addition of the second transaction to the blockchain, while the first transaction is discarded. The outcome of this malicious act is that the attacker is able to spend the same digital asset twice, resulting in financial losses for the victims of the attack. To mitigate the threat of double spending attacks, blockchain networks use consensus mechanisms and validation processes that verify each transaction's uniqueness and ensure that digital assets can only be spent once (*Malik et al., 2019*).

### Cracking of the cryptographic

The term "cryptographic cracking" pertains to the act of bypassing or breaking the security measures implemented by cryptographic algorithms. Cryptography is a technique employed to safeguard digital data by converting it into an indecipherable format using encryption methods. Breaking a cryptographic algorithm involves identifying weaknesses or loopholes in the encryption process and exploiting them to gain entry to the encrypted information. This could result in unauthorized access to sensitive data, such as personal data or financial records, leading to severe security breaches. To prevent cryptographic cracking, developers employ robust cryptographic algorithms and encryption techniques that are difficult to compromise, and regularly update and enhance their security measures to stay ahead of potential attacks. With the utilization of cryptographic techniques in

blockchain to secure data, it is currently nearly impossible to crack cryptographic algorithms with the aid of CC attacks (*Moubarak, Filiol & Chamoun, 2018*).

### Denial of Service (DoS) attack

Denial of Service (DoS) attacks can have a serious impact on the normal functioning of a blockchain network by overwhelming it with a flood of traffic or requests. This type of attack can be carried out through various means, including transaction flooding, Distributed Denial of Service (DDoS) attacks, and resource depletion. To protect blockchain networks from such attacks, security protocols such as firewalls, load balancers, and anti-DDoS software can be implemented. Additionally, it is important to limit the number of transactions that can be processed per second. Some blockchains have implemented consensus mechanisms, such as Proof of Stake (PoS) and Proof of Authority (PoA), to make it more difficult for attackers to overwhelm the network (*Zhong & Guo, 2021*; *Mirkin et al., 2020*). There are different types of DoS attacks in blockchain technology, including:

**Transaction flooding:** An attacker sends a large number of transactions to the network, overloading the nodes and causing delays in transaction processing.

**Distributed Denial of Service (DDoS) attack:** This type of attack involves multiple attackers targeting the same network with a large volume of traffic, making it difficult for the network to function properly.

**Resource depletion:** An attacker consumes a large amount of resources, such as storage space or bandwidth, which can cause the nodes to crash or slow down (*Yatsykovska et al., 2011*).

### Sybil attack

Sybil attacks in blockchain technology refer to the act of an attacker creating multiple fake identities or nodes to gain control over the network. This attack can have severe consequences, including the ability to manipulate transactions, prevent certain transactions from being processed, change the blockchain's history, overwhelm the network with fake transactions or requests, and even perform double-spending attacks. To prevent Sybil attacks, blockchain networks can implement various measures, including Proof of Work (PoW), Proof of Stake (PoS), and identity verification. These measures make it more difficult for an attacker to create multiple fake identities and gain control over the network. Overall, Sybil attacks are a serious threat to blockchain networks, and it is important to implement appropriate security measures to prevent them. The choice of consensus mechanism and other security measures should be carefully considered to protect against such attacks (*Platt & McBurney, 2021*; *Zhong & Guo, 2021*).

### WannaCry ransomwarecrypto-worm attack

The WannaCry ransomware attack occurred in May 2017 and targeted computers running Microsoft Windows operating systems. It aimed to encrypt files on infected computers and demand payment in Bitcoin for the decryption key. This attack affected over 230,000 computers in more than 150 countries and caused significant disruption, particularly in the healthcare sector (*Whittaker, 2019*).

### Petya attack

The Petya ransomware attack, like the WannaCry attack, specifically targeted computer systems using Microsoft Windows as their operating system. The aim of the attack was to infect the master boot registry and encrypt the hard drive's file system, preventing the system from booting into Windows. Similar to WannaCry, this attack demanded a ransom payment in Bitcoin for the decryption key (*Singh & Singh, 2016*).

### Transaction malleability attack

A transaction malleability attack is a type of attack where an attacker can manipulate a transaction ID, tricking the user into thinking that a transaction was not completed and prompting the user to repeat the transaction with additional payments. This can result in the user unknowingly paying twice for the same transaction. Essentially, the attacker plays a game with the user, forcing them to act in a particular way that benefits the attacker. This attack is typically accomplished through hacking the transaction ID (*Decker & Wattenhofer, 2014*).

### Timejacking

A timejacking attack in blockchain technology is a sophisticated type of attack that specifically targets the timestamp of the blockchain. In this type of attack, the attacker manipulates the time counter of a node to trick it into using an alternative or fake blockchain. By doing so, the attacker can create confusion and potentially manipulate the blockchain's history, leading to disastrous consequences (*Moubarak, Filiol & Chamoun, 2018*). This type of attack can be particularly dangerous in proof-of-work blockchain systems where the validity of a block is determined by its timestamp. As a result, it is essential to prevent timejacking attacks. Some blockchain systems have mechanisms in place, such as checkpointing, which involves pre-determined checkpoints that nodes can use to verify the blockchain's history and detect any malicious activity. Furthermore, implementing secure time synchronization protocols can also help prevent timejacking attacks and ensure that all nodes have synchronized and accurate time counters.

### Routing partition attack

A routing partition attack in blockchain technology is a type of attack where the attacker captures and modifies data or transactions between nodes before they are transmitted to other peers in the network. The attacker achieves this by breaking the entire network into small groups of nodes that are unwilling to communicate with each other, ensuring that the attacker's malicious activity remains undetected. This type of attack can have serious consequences as it can allow the attacker to manipulate the blockchain's history and potentially steal cryptocurrency. To prevent routing partition attacks, blockchain systems can implement measures such as multi-path routing, where transactions are transmitted through multiple paths to ensure their integrity and prevent interception by attackers. Additionally, implementing secure communication protocols between nodes and regularly monitoring the network for any signs of suspicious activity can also help prevent routing partition attacks (*Singh & Singh, 2016*).

### Delay/jellyfish attack

A delay attack, also known as a jellyfish attack, guaranteed time slot (GTS) attack, or timing attack, is a type of attack in blockchain technology that aims to disrupt the network by causing delays in the transmission of data packets. The attacker achieves this by propagating the data packets through devices, which results in unnecessary delays and can prevent the blockchain from functioning properly (*Zhong & Guo, 2021*). This type of attack can be particularly dangerous in real-time blockchain systems, where timing is critical to ensure the proper functioning of the network. To prevent delay attacks, some blockchain systems use mechanisms such as priority scheduling, which prioritizes the transmission of data packets based on their importance, or bandwidth reservation, which allocates a guaranteed amount of bandwidth for critical data packets. Additionally, implementing secure time synchronization protocols can also help prevent delay attacks.

### Eclipse attack

This type of attack is known as an IP address spoofing attack in blockchain technology. In this attack, the attacker gains control of a large set of IP addresses, often through the use of a distributed botnet. When the victim restarts its system or blockchain, the connection is reset, and the attacker-controlled IP addresses are able to intercept the data or transactions that are sent. By spoofing the IP address of the victim, the attacker can make it appear as though the data or transactions are coming from a trusted source, potentially causing significant damage to the blockchain system. To prevent IP address spoofing attacks, some blockchain systems use techniques such as packet filtering and access control lists to verify the authenticity of incoming data packets. Additionally, implementing secure communication protocols such as Transport Layer Security (TLS) can also help prevent IP address spoofing attacks (*Singh & Singh, 2016*).

### Phishing

Phishing represents a form of cryptocurrency scam where perpetrators deceive victims into divulging their private keys or personal details. The attacker often adopts a false identity, pretending to be a genuine individual or entity to establish trust with the victim. Once the victim falls prey to the scheme, the attacker exploits the obtained information to pilfer their cryptocurrency funds. Phishing starts with a mass email or message from the attacker, appearing legitimate, with a link to a fake website resembling the real one. When victims input their login info on the fake site, the attacker gains access to their account. In 2022, "Malicious browser bookmarks," "Zero dollar purchase," "Trojan horse currency theft," "Blank Check", and "Same ending number transfer scam" emerged as notable examples of phishing attacks (*Katte, 2023*). Spear phishing attack, Whaling attack, Clone phishing attack, Pharming attack, Evil twin attack, Voice phishing attack, SMS phishing attack, and so on are some well known phishing attacks in blockchain (*Katte, 2022*).

### Vulnerable signatures

In blockchain technology, the vulnerable signatures attack is a specific type of security vulnerability that targets the signature mechanism used for authentication and verification of transactions. The attacker intercepts the digital signature, replicates it, and uses it to

target multiple victims simultaneously. This leads to unauthorized access, data theft, and other malicious activities. To mitigate the risk of vulnerable signature attacks, blockchain systems use secure signature algorithms and cryptographic protocols that are resistant to forgery and tampering. Moreover, frequent security audits and updates can help detect and address vulnerabilities in the signature mechanism, ensuring the overall integrity and security of the blockchain system.

### Dictionary attacks

The dictionary attack constitutes a form of brute-force attack utilized to illicitly access confidential information or systems, including passwords, hashes, digital signatures, and encryption algorithms. In this attack strategy, the assailant adopts a hit-and-trial methodology, commencing with a repository of commonly employed passwords and personal details, such as names and dates of birth. Subsequently, the attacker meticulously endeavors each entry from the compiled list until the correct password or sensitive information is successfully identified. The objective of this attack is to exploit the vulnerabilities of weakly guarded passwords and authentication mechanisms, thereby gaining unauthorized entry to the targeted system or compromising the security of cryptographic elements (*Tosh et al., 2017*; *Houy, Schmid & Bartel, 2024*). It is imperative to be aware of the dictionary attack's *modus operandi*, as it underscores the significance of employing robust security measures to safeguard against such malicious endeavors. In the context of blockchain-based systems, the term "dictionary attack" is not commonly used as it would be in the context of traditional password-based systems. Blockchain systems typically rely on cryptographic keys and signatures, and brute-force attacks like dictionary attacks are not feasible due to the extremely high computational effort required to break the cryptographic algorithms. Instead, in the context of blockchain security, the focus is on protecting private keys, preventing unauthorized access to wallets or accounts, and securing the consensus mechanism. Threats in blockchain-based systems are more likely to involve attacks on the underlying protocols, vulnerabilities in smart contracts, or social engineering techniques to trick users into revealing their private keys.

### Flawed key generation

The flawed key generation attack is a type of attack that aims to hack private keys used for authentication and verification in a system. This attack takes advantage of the user's negligence or lack of knowledge, as they fail to update the contents of the keys in a safe and secure manner. As a result, the keys become vulnerable to attacks, which can lead to unauthorized access, data theft, and other malicious activities. To prevent flawed key generation attacks, users should regularly update and secure their private keys using strong encryption and authentication methods, and follow best practices for key management and storage.

### Attacks on hot block

An attack on the hot block in blockchain technology refers to a security breach that targets the storage of private keys used in cryptographic techniques. Hot block refers to the storage of private keys in an application that is connected to the internet, making it vulnerable to

attacks. Attackers can exploit vulnerabilities in the application to gain unauthorized access to the private keys and use them to perform various malicious activities, such as stealing funds or manipulating transactions (*Moubarak, Filiol & Chamoun, 2018*).

**(i) Definition of attack strategy:** Due to its decentralized nature, the blockchain ensures the anonymity of its users. However, users are unaware of whether the blocks contain incorrect transaction information. In contrast, attackers have the capability to access information about the state of the block. The attacker can selectively join a newly created block that contains fraudulent transaction information behind a block that aligns with their interests. The selection strategy employed by the attacker can be defined as follows:

$$R \begin{cases} \max\limits_{V_i \in U_{attack}} \sum\limits_{i=1}^{n} child\,(V_i), \sum\limits_{i=1}^{n} V_i \geq 1 \\ \max\limits_{V_i \in U_{all}} \sum child\,(V_i), \sum\limits_{i=1}^{n} V_i = 0 \end{cases} \tag{3}$$

R, denotes the current linked block in the blockchain that the attacker selects, with each block representing a node in the blockchain. The "child()" function is utilized to determine whether the node has any children. In the presence of an attack block within the blockchain, the attacker connects the new block to the longest chain after the attack block. If no attack block exists, the attacker selects the relatively longest branch to connect the new block.

**(ii) Honest miner strategy definition:** For an honest miner, the type of block remains unknown. In a blockchain, the system only recognizes transaction information in the block with the longest chain. At the probability level, honest miners can connect newly generated nodes to any block, but they will ultimately choose the longest leaf node. If multiple leaf nodes belong to the same chain length, the attacker will randomly connect to one of the leaf nodes with equal probability. As the node depth decreases by one layer, the probability of a node being selected will decrease by half. The sum of the probabilities of all selected nodes in Eq. (4) equals one. Equation (5) is utilized to determine the relationship between the probability of selecting a node and the level of the tree in which it is located.

$$\sum_{i=1}^{n} \sum_{j=1}^{m} (1/2)^{(L-i)} p = 1 \tag{4}$$

$$p_{i_j} = (1/2)^{(L-i)} p \tag{5}$$

where, L represents the length of the entire blockchain, and the probability of selecting a leaf node as the longest chain is denoted by p, which is determined by the current state of the blockchain. In practical operating environments, the probability of an honest miner selecting a node before reaching a leaf node is relatively low (*Zeng et al., 2019*). Honest attackers adopt a more sophisticated strategy compared to other attackers, as they lack knowledge of the state of each block. As the proximity to the root node increases, the probability of a node being selected decreases.

## Transparency

The characteristics of distributed bookkeeping, complete replication, and traceability make it possible to track and query all interactions in blockchain. This leads to increased transparency of information interaction. However, despite its potential advantages, blockchain technology is a complex system that involves many disciplines such as cryptography, artificial intelligence, and computational mathematics (*Garay, Kiayias & Leonardos, 2015*). Currently, there is a lack of information talent in some cross-cutting areas of cutting-edge technology. Furthermore, the large-scale commercial application of blockchain is hindered by a series of limitations such as the energy-consuming consensus mechanism for proof-of-work, limited block capacity, and long confirmation time. Additionally, privacy disclosure caused by data transparency and seamless connection with already existing systems, as well as legal and regulatory issues, require ongoing research and resolution (*Cui, 2022*).

## Blockchain volume issues

In the blockchain system, each network node retains all the data records on the blockchain. However, as the blockchain grows rapidly, the amount of data stored in each node increases, resulting in a heavier computational & storage burden. For a new user to participate in reviewing and tracking specific transactions in the blockchain network, they must first spend a certain amount of time and storage space to load all the block records on the blockchain. In the case of the Bitcoin blockchain, the complete data volume by 2019 exceeds 70 GB, and it can take more than 3 days for a new user to synchronize the data using the core Bitcoin client after joining the Bitcoin network (*Liu & Zou, 2019*). According to a web blog the data volume by 2023 has already crossed 500 GB (*de Best, 2023*).

# INTRODUCTION TO FUTURE RESEARCH METHODS AND APPLICATION AREAS

The development and application of new information technology are accelerating as the development of blockchain accelerates in terms of policy, technology, and the environment in which it is applied. Countries are making unprecedented efforts to gain a technological advantage in the frontier areas of the future. The competition among nations is becoming more complex and intense. Blockchain technology, which vigorously promotes economic and industrial restructuring, can be a critical turning point for developing countries to achieve leapfrog progress and occupy an unshakable position in the international division of labor. Financial research institutions are paying more attention to the technology based on blockchain "distributed bookkeeping," with its programmable, distributed, chronological, encrypted, and tamper-proof technology (*Huckle et al., 2016*).

## Characteristics of beneficial systems

Systems that can benefit from blockchain usually have many participants who do not fully trust each other but need to work together transparently and securely. They involve frequent transactions or data transfers, and they may require unique digital identifiers,

decentralized naming services, and secure ownership systems. These systems may also want to reduce manual efforts in solving problems and disputes, and they need to allow real-time monitoring by regulators. Having a complete record of all transactions and assets is also essential.

## Suitable applications for blockchain technology

Blockchain is great for systems that need secure, transparent, and decentralized ways of handling transactions or data. For example, it can be helpful in supply chain management (*Ahmed, 2022*; *Nethravathi et al., 2022*), financial systems, healthcare data management, digital identity systems, voting, and real estate transactions (*Levy, 2022*; *Zheng et al., 2018b*; *Dai, Zheng & Zhang, 2019*). There are plethora of blockchain applications in our daily lives we have mentioned here a few as follows.

### Agricultural quality and safety traceability system

The application of blockchain technology can record all production and distribution data, enabling consumers to track the production and transportation records of agricultural products throughout the process. The primary technical features of blockchain, including distributed storage, decentralized management, shared maintenance, consensus trust, and reliable database, can be utilized to provide solutions to the quality problems of agricultural products (*Hua et al., 2018*; *Srivastava, Zhang & Eachempati, 2023*).

### Education

In October 2016, the Ministry of Industry and Information Technology published a "White Paper on the Development of Blockchain Technology and Applications," which highlighted that the transparency and immutability of data in the blockchain system can be utilized for student credit management, graduate employment, academic research, industry-university cooperation, and qualification certification. This is of great significance for promoting the healthy development of education and employment (*Budiharso & Tarman, 2020*). The development of a decentralized education system using blockchain technology can help break the monopoly of education or government agencies on the right to education. It can create a comprehensive education system in which all parties can participate and coordinate construction. In the future, not only schools, training units, and other educational institutions approved by government departments with qualifications to provide educational services but also more institutions and even individuals can assume the role of professional educational service providers. Moreover, the open-source nature, transparency, and tamper-proof of blockchain can guarantee the authenticity and credibility of the educational process and results (*Terzi et al., 2021*; *Yin et al., 2022*).

### The convergence of IoT and blockchain

The Internet of Things (IoT) is an extension of the internet that connects various devices and objects, enabled by computer network technology and employing smart chips, RF devices, and communication modules for information sharing and automated identification of goods. It represents the latest wave in the information industry following the advent of computers, mobile communication networks, and the internet. IoT systems

collect and transmit information about objects wirelessly, creating a vast data analysis system for use in various fields, including intelligent transportation, manufacturing, power grids, and homes. Despite the various application scenarios of IoT, some problems must be addressed before the technology can mature, such as the challenges associated with maintenance costs, cloud services, and vulnerable devices susceptible to attacks that cause data loss and high maintenance costs.

To overcome these challenges, blockchain technology can help facilitate the large-scale commercialization of IoT. Blockchain can replace the role of a central server by involving multiple nodes in verifying and recording transactions across the network in a distributed ledger. The blockchain consensus mechanism allows all network nodes to be verified, and the use of asymmetric encryption technology and distributed data storage significantly reduces the risk of hacker attacks. By combining the two technologies, practical problems in the physical world can be effectively solved. Furthermore, blockchain technology can provide protection for information security and other issues related to the Internet of Things.

### Energy

The energy industry is undergoing a shift towards a clean and distributed approach in response to the energy revolution and environmental protection movement, resulting in a new energy structure with complementary energy flows. The bottom-up distribution of the energy system will effectively complement the traditional energy system. In this process, blockchain technology is likely to become an important means to realize the infrastructure of the Internet of Energy (IoE). By combining distributed trading systems and clean energy in the energy industry, blockchain can popularize these two trends and promote their widespread use. Blockchain can improve the efficiency of energy production, enhance monitoring accuracy, reduce management costs, and secure the wholesale energy trading market. It can also reduce communication costs, promote the development of clean energy, provide timely payment and settlement systems for retail energy trading markets, increase investment and financing channels, and reduce energy investment and financing risks. These measures can increase participation, provide liquidity for energy saving and emission reduction in the energy sector, and help achieve the goal of stabilizing climate change. One of the most promising applications of blockchain in energy management is energy trading. By using blockchain, energy producers and consumers can trade energy directly with each other, without the need for intermediaries, thus reducing transaction costs and increasing the efficiency of the energy market (*Münsing, Mather & Moura, 2017*). Blockchain can also improve grid security by providing a tamper-proof and transparent platform for recording energy transactions and ensuring the integrity of energy data. This can help prevent cyber attacks and ensure the reliability of the energy grid (*Bergquist et al., 2017*). Furthermore, blockchain technology can be used for electricity market control, enabling more efficient management of energy resources and reducing waste. By automating the process of matching supply and demand and providing real-time data on energy production and consumption, blockchain can help balance the energy grid

and reduce the need for traditional energy providers (*Lundqvist, de Blanche & Andersson, 2017*).

### Digital identity

With the rapid development of the Internet, digital identity is becoming increasingly prevalent in various industries. Generally, digital identity enables the association of a person's stored computer information with their societal identity. Broadly speaking, a digital identity is used to identify an individual's presence in an Internet scenario and is a combination of relevant characteristics. Digital identity can represent physical information about an external agent, such as an individual, a business, or a government, through a computer system. Digital identity can create a better and more trustworthy environment for the Internet and is a fundamental basis for the digitization of financial transactions worldwide. Blockchain technology can be a viable solution to some of the issues related to digital identity, such as privacy concerns and data sovereignty. Blockchain can prevent the use of false information through unilateral use, such as phone numbers or address information, which helps avoid identity theft and eliminates the risk of inconsistent information resulting from the use of personal digital identities in various contexts. Moreover, blockchain technology uses asymmetric cryptography in the verification phase, verifying the identity of the requester by comparing the hash values of digital identities without the original data, thus eliminating the risk of personal privacy leakage.

### IoT

The blockchain technology has several promising applications in the IoT domain. It can facilitate secure and decentralized data sharing among IoT devices without intermediaries, provide a secure identity and authentication mechanism, enable tracking and tracing of products in the supply chain, automate and enforce contracts between IoT devices using smart contracts, and facilitate peer-to-peer energy trading. As research and development in this area continues, it is likely that additional use cases for blockchain in IoT will emerge, making it an important technology for enabling secure and autonomous IoT networks. IoT applications need trust mechanisms that ensure the integrity of the collected data and the associated interactions as well as their transparency, that blockchain can provide (*Sicari et al., 2015*). The research community puts a lot of interest in the integration of blockchain into different aspects of IoT—decentralization (*Veena et al., 2015*), security (*Khan & Salah, 2018*), anonymity (*Christidis & Devetsikiotis, 2016*), identity (*Gan, 2017*), and device management (*Samaniego & Deters, 2016*).

### Finance

The potential of blockchain technology in the finance sector is high and widely recognized (*Peters & Panayi, 2016*). Research efforts are focused on enhancing transaction processing speed and performance (*Peters & Panayi, 2016*), as well as strengthening security and data privacy (*Singh & Singh, 2016*). Furthermore, blockchain is being explored for its ability to automate financial contracts (*Egelund-Müller et al., 2017*) and support corporate finance (*Momtaz, Rennertseder & Schröder, 2019*), among other applications.

### Healthcare

Blockchain technology has numerous applications in the healthcare sector, making it a promising technology for revolutionizing the industry (*Ramzan et al., 2023*; *Gupta et al., 2022*). One of the primary applications of blockchain in healthcare is electronic medical records (EMRs) management. By leveraging blockchain's decentralized and secure nature, EMRs can be shared securely and efficiently among healthcare providers, improving patient outcomes and reducing costs (*Gordon & Catalini, 2018*). Another promising application of blockchain technology in healthcare is biomedical research. Blockchain can be used to facilitate secure and transparent sharing of research data among researchers and institutions, ultimately accelerating the pace of medical discoveries (*Benchoufi, Porcher & Ravaud, 2017*). Blockchain can also be used for drug supply chain management, which is critical for ensuring the authenticity and safety of pharmaceuticals. By recording every step in the supply chain on a blockchain, stakeholders can track the movement of drugs from manufacturers to patients, reducing the risk of counterfeit or contaminated drugs (*Tseng et al., 2018*). Furthermore, blockchain technology can streamline insurance claim processing, allowing for faster and more accurate claims processing, reducing fraudulent claims, and improving overall transparency and efficiency (*Zhou, Wang & Sun, 2018*). Overall, the wide-ranging applicability of blockchain in the healthcare industry shows great potential for improving patient outcomes, enhancing data privacy, and reducing costs.

### Government

Blockchain technology has the potential to revolutionize the government sector by enhancing transparency, security, and efficiency in a variety of areas. One of the most promising applications of blockchain technology in government is e-government services. By leveraging blockchain's decentralized and secure nature, governments can offer secure and efficient digital services, such as the issuance of licenses, permits, and certificates (*Batubara, Ubacht & Janssen, 2018*). Another critical application of blockchain technology in government is digital identity management. Blockchain-based identity systems can provide a tamper-proof and transparent platform for managing personal data, enabling secure and efficient digital identity verification and reducing the risk of identity theft (*Dunphy & Petitcolas, 2018*). Blockchain can also be used to enhance the integrity and transparency of e-voting systems, enabling secure and transparent voting processes that are resistant to tampering and fraud (*Pawlak, Guziur & Poniszewska-Marańda, 2019*). Furthermore, blockchain-based value registries can provide a secure and transparent platform for recording and tracking the ownership and transfer of assets, such as land titles or intellectual property rights (*Ramya et al., 2019*). Overall, the use of blockchain technology in government has the potential to improve transparency, reduce corruption, and enhance the efficiency and security of public services.

### Artificial intelligence (AI)

The combination of blockchain and AI technologies offers a range of potential benefits across various fields (*Ogundokun et al., 2022*). One of the most promising applications of

this synergy is the ability to track the provenance of AI training models, enabling greater transparency and accountability in the development and deployment of AI systems (*Sarpatwar et al., 2019*). In addition, the integration of blockchain and AI can enhance the efficiency and security of transportation systems. By leveraging blockchain's decentralized and secure nature and AI's predictive capabilities, transportation networks can be optimized for faster, more efficient, and safer operations (*Yuan & Wang, 2016*; *Singh et al., 2022*). Furthermore, the use of blockchain and AI can improve the control of robots, enabling more sophisticated and autonomous decision-making capabilities. This can enhance the efficiency and effectiveness of a range of industries, from manufacturing to healthcare (*Lopes, Alexandre & Pereira, 2019*). The synergy of blockchain and AI can also support the development of IoT networks, providing a secure and transparent platform for managing the vast amounts of data generated by IoT devices. This can enable more efficient and effective management of IoT networks, enhancing their potential to revolutionize various industries (*Singh, Rathore & Park, 2020*). Overall, the integration of blockchain and AI has the potential to enhance the efficiency, security, and transparency of a range of industries, offering exciting possibilities for the future of technology.

### Decentralized AI

Decentralized Artificial Intelligence (DAI) represents a revolutionary AI system that harnesses the power of Blockchain technology to store and process data. Unlike centralized AI systems controlled by a single authority, DAI relies on consensus among multiple nodes, ensuring a more secure, transparent, and trustworthy approach to decision-making (*Adel, Elhakeem & Marzouk, 2022*; *Rana et al., 2022*). The prevalence of AI has surged in recent years, with organizations increasingly adopting AI systems. The average number of AI systems per organization has doubled (*Sharma, 2023*) from 1.9 in 2018 to 3.8 in 2022. While AI offers enhanced technologies and solutions across industries, its implementation can be costly, potentially leaving some behind in the digital divide. However, AI's potential in Metaverse development is significant (*Hwang & Chien, 2022*; *Cao, 2022*). Centralized AI systems face critical challenges due to their dependence on large data sets, raising concerns about democratizing data and intelligence or retaining control within a few organizations. To address these challenges, decentralized AI powered by Blockchain technology emerges as a solution. Decentralized AI projects benefit from the openness and traceability of shared ledgers, making them publicly verifiable by anyone. Platforms like SingularityNet enable smaller companies to offer AI applications as a service, democratizing market access for startups. Decentralization also fosters increased innovation by allowing multiple entities to contribute to AI system development and decision-making. This diversity of perspectives leads to a wider range of ideas and advancements in the AI landscape (*Cao, 2022*).

### Big data

Blockchain technology has significant potential to address some of the key challenges in the field of big data. By leveraging blockchain's decentralized and secure nature, a data-sharing platform can be established that enables the secure and efficient exchange of data

among all involved parties (*Chen & Xue, 2017*). In addition, the use of blockchain can improve the reliability of big data by ensuring the integrity and accuracy of the data shared between parties (*Abdullah, Hakansson & Moradian, 2017*). This can enhance the trustworthiness of big data analytics and enable more informed decision-making processes. Moreover, blockchain technology can increase the security of big data by providing robust cryptographic protection against tampering, hacking, and other security threats. This can help to prevent unauthorized access to sensitive data, safeguarding it from malicious attacks. Furthermore, the use of blockchain can provide timestamping capabilities, enabling the creation of an immutable record of all transactions and interactions between parties. This can enhance the auditability and traceability of big data, enabling greater transparency and accountability. Overall, the use of blockchain technology in big data has the potential to improve data sharing, reliability, security, and timestamping, enabling more efficient and effective management of big data analytics.

### Money transfers

Blockchain enables fast cross-border transactions, completing within minutes compared to traditional transfers taking days with high fees and intermediaries (*Hashemi Joo, Nishikawa & Dandapani, 2020*). By eliminating intermediaries like banks, blockchain reduces transaction costs, making transfers more affordable. Its decentralized and encrypted nature ensures secure transactions, with each transaction recorded transparently and immutably, preventing fraud. Users can track fund flow in real-time, enhancing trust and accountability.

### Lending

Blockchain smart contracts automate lending, eliminating intermediaries like banks. Borrowers directly interact with the contract, which executes loan terms automatically when conditions are met, streamlining the process and reducing overhead (*Chen et al., 2018*). Smart contracts operate on a transparent and immutable blockchain ledger, ensuring transparency and trust for all parties. Borrowers and lenders have complete visibility into loan terms, and execution is verifiable by stakeholders. Blockchain's decentralized nature reduces fraud risk, and cryptographic techniques ensure data privacy during lending. By eliminating intermediaries and automating processes, blockchain smart contracts significantly reduce lending operational costs, benefiting borrowers and lenders. Smart contracts expedite loan approvals by automating verification steps, benefiting borrowers in urgent financial situations. Smart contracts manage collateral for secured loans, automatically releasing it to borrowers or transferring ownership to lenders when conditions are not met. Smart contracts accommodate various loan terms, customized to suit borrowers' needs. Blockchain smart contracts enable borderless lending, offering opportunities worldwide without intermediaries.

### Insurance

Financial insurance encompasses a wide array of activities, from stock trading and equity management to bonds, fundraising, inter-institutional clearing and settlement, fund management, and the issuance of insurance certificates. However, maintaining the

authenticity of financial interests and ensuring secure transactions between untrusted parties has been a persistent challenge in this sector (*Weber, 2010*). For example, within insurance management, trust is a fundamental element, and traditional social trust mechanisms often fall short of meeting the evolving needs of the social economy. To address this, the integration of blockchain technology, encryption, distribution, and related identification technology offers a comprehensive framework and solution for the unique identity challenge. This integration ensures the authenticity and reliability of information and data, while also providing effective source tracking and robust technical support for preventing insurance fraud. By combining blockchain's timestamping and distributed features with IoT (Internet of Things) technology, we have the opportunity to solve the insurance sector's problems. Moreover, blockchain facilitates inter-temporal information management and encourages innovation in insurance products and services, particularly in fragmented scenarios. This innovation enables the development of more detailed and flexible insurance solutions tailored to specific risk objectives. As a result, it supports the implementation of personalized and efficient insurance services. The introduction of blockchain, coupled with smart contracts, offers significant advantages to the insurance sector. Smart contracts enable the transparent recording of claims on an immutable ledger, reducing the risk of duplicate claims (*Raikwar et al., 2018*). This, in turn, leads to faster claims processing, ensuring swift compensation for customers. The decentralized nature of blockchain technology provides robust security for sensitive data, fostering trust between customers and insurance providers (*Saldamli et al., 2020*). This technology-driven approach revolutionizes insurance operations, delivering transparency, expediting settlements, and enhancing security. Ultimately, it paves the way for a more sustainable and efficient insurance industry (*Crawford, 2017*).

### Secure personal information

The utilization of blockchain technology to secure and manage personal identifying information, such as Social Security numbers and date of birth, presents a compelling approach to address the security vulnerabilities prevalent in conventional centralized systems. By leveraging the decentralized and immutable nature of blockchain, the risk of data breaches and unauthorized access can be substantially mitigated (*Takemiya & Vanieiev, 2018*; *Hakak et al., 2020*).

### Voting

If we store personal identity information on a blockchain, it brings us closer to the possibility of using blockchain for voting. Blockchain technology ensures that nobody can vote twice, only eligible voters can participate, and no one can alter votes. Additionally, it makes voting more accessible by allowing people to vote easily through their smartphones with just a few taps. Using blockchain for voting would also reduce the cost of conducting elections significantly (*Yavuz et al., 2018*; *Shahzad & Crowcroft, 2019*; *Hanifatunnisa & Rahardjo, 2017*).

### Proof of ownership

The concept of proof of ownership encompasses the establishment of ownership for a wide spectrum of assets, spanning both physical and intangible realms. This spectrum includes property certificates, patents, trademarks, and copyrights, with copyrights being of particular significance. Copyrights grant authors of literary, artistic, and scientific works a combination of personal and property rights over their creations. However, in the face of the expanding content industry, intellectual property infringement has surged as a critical concern. Despite the presence of government policies and regulations aimed at safeguarding intellectual property rights, piracy persists as an enduring problem, resulting in substantial economic losses for the rightful owners of such assets (*Dramé-Maigné et al., 2018*; *Crosby et al., 2016*). In addressing this persistent issue of piracy, blockchain technology emerges as an ideal solution due to its seamless integration with digital copyright protection systems. Significantly, the current process of confirming copyright protection is often marred by its time-consuming and inefficient nature, a challenge that blockchain technology is well-equipped to overcome. Leveraging distributed ledger and timestamp technologies inherent in blockchain expedites the consensus-building process concerning intellectual property ownership and associated rights. Consequently, blockchain technology offers a promising avenue for timely copyright confirmation, mitigating the inefficiencies inherent in the present system. Additionally, asymmetric encryption technology ensures the uniqueness of copyright, while timestamping provides a clear and indisputable means of identifying copyright ownership. As a result, copyright owners can efficiently and promptly verify their rights, ultimately leading to a more effective system of copyright protection.

Furthermore, blockchain technology extends its transformative potential to support artists in securing their intellectual property rights on the Internet. By harnessing the decentralized and immutable nature of blockchain, artists can establish a transparent and tamper-proof record of their creative works, granting them heightened control and security over their intellectual property rights. Notably, blockchain's innate capability to prevent the duplication of files positions it as a potent tool in combating piracy within the digital domain. The implementation of blockchain to track playbacks on streaming services, coupled with the utilization of smart contracts for payment distribution, promises artists greater transparency and assurance in receiving fair compensation for their creative endeavors. This multifaceted approach underscores the transformative impact that blockchain can have in empowering artists and safeguarding their rights in the digital age (*Gürkaynak et al., 2018*; *Wang et al., 2019a*; *Tsai et al., 2017*; *Singh & Tripathi, 2019*).

### Maritime supply-chain management

As the world continues to globalise, international trade and logistics are expanding rapidly. Marine logistics has long been a crucial economic mode of transportation. Technological advances, such as the Internet of Things and high-performance big data analysis, have led to the proposal of numerous blockchain-based solutions for big data analysis (*Jiaguo Liu & Zhen, 2023*).

# LIMITATIONS OF BLOCK CHAIN TECHNOLOGY

Blockchain technology has some important limitations and misconceptions also (*Yaga et al., 2018*; *Malik et al., 2019*; *Shen et al., 2022*; *Taherdoost, 2022*) and also by *Biswas (2023)*.

## Interoperability

The diversity in protocols, algorithms, and data structures across various blockchains hinders seamless information exchange, limiting their potential as universal transaction platforms. For example, blockchains like Bitcoin and Ethereum lack meaningful communication capabilities, impeding complex applications. This is exacerbated by disparate programming languages for their smart contracts, necessitating dual proficiency for developers. The absence of interoperability leads to high transaction fees and constrains multi-network applications, curbing the broader adoption of blockchain technology. Initiatives to foster data transfer between blockchains are emerging, but interoperability remains a significant challenge (*Belchior et al., 2021*).

## Distributed nature

A Blockchain functions by dispersing data across a network, forming an uninterrupted series of records resistant to tampering and modification. This architecture integrates blocks containing data or executable programs. Each block aggregates discrete transactions and the results of executed blockchain operations. The bedrock of trust within the blockchain framework emanates from the widespread presence of a complete chain replica, meticulously recording every transaction, consistently upheld throughout the network. However, managing this decentralized system, characterized by participants spanning numerous computers, can prove intricate, particularly concerning consensus and maintaining synchronization among all stakeholders (*Patel et al., 2020*).

## Private keys

The security of the blockchain network is primarily upheld by the concept of private keys. These private keys play a crucial role in validating blockchain addresses and ensuring the integrity of transactions. When a user opens a cryptocurrency wallet, they are provided with a unique private key, which essentially serves as a password granting access to withdraw funds from the wallet. Losing the private key can be catastrophic, as it renders the user unable to access their funds. To mitigate this risk, it is essential to store multiple copies of the private key securely. This way, if the original key is lost or compromised, the user can still rely on one of the backup copies to regain access to their wallet. However, the practice of maintaining multiple copies of the private key also introduces a potential vulnerability. If unauthorized individuals gain access to any of these copies, the entire crypto wallet becomes compromised, exposing the user's assets to theft or misuse (*Malik et al., 2019*). Unlike typical passwords used for social media or email accounts, private keys cannot be changed once they are generated. This lack of flexibility in altering private keys further emphasizes the need for utmost caution in their storage and protection.

## No trusted third party

Trusted third parties such as banks, institutions, and governmental bodies, despite regulatory oversight, still have vulnerabilities—banks can collapse, work providers can face insolvency, and officials can exhibit corruption. In contrast, the blockchain presents a practical alternative, functioning autonomously and obviating the requirement for intermediaries, thereby guaranteeing dependable execution without dependence on conventional trusted institutions (*Halaburda, 2018*; *Gamage, Weerasinghe & Dias, 2020*). However, while the trustless nature of the blockchain offers advantages, it also introduces uncertainties, particularly in cases of transaction failures where traditional banks assume responsibility, raising questions about accountability in blockchain-based systems (*Bitstamp Learn, 2022*).

## High cost

The adoption of blockchain technology entails significant financial investments, making it a capital-intensive endeavor for most companies. This financial barrier serves as a deterrent to many enterprises considering the implementation of blockchain solutions. Company owners seeking to incorporate blockchain into their operations must be prepared for substantial expenses (*Zhang et al., 2020a*; *Alammary et al., 2019*). One of the primary cost components is the need to hire proficient and specialized personnel. This includes hiring core blockchain developers and blockchain software developers who possess the expertise to design, build, and maintain blockchain systems. Given the scarcity of skilled professionals in this domain, the cost of acquiring such talent can be substantial. Additionally, the development of blockchain-based applications further adds to the financial burden. Companies must allocate resources to create applications that leverage blockchain technology effectively, tailored to meet their specific needs and requirements. Moreover, the hardware infrastructure necessary to support blockchain networks contributes to the overall expenses. The robust and decentralized nature of blockchain demands sophisticated hardware setups capable of maintaining the integrity and security of the distributed ledger.

## Transactional workflow

The inherent distribution in blockchains renders them especially well-suited for inter-organizational e-Business applications (*Lokshina, 2022*). By cryptographically endorsing blocks housing transactions, blockchains establish an immutable record. Within a distributed blockchain, participants create a peer-to-peer (P2P) network to autonomously verify transactions and integrate them into the blockchain. In the context of inter-organizational workflow management, consensus among participants is pivotal to determine work status, influencing the array of subsequent valid actions in the process. However, while blockchain is tailored for high-frequency transactions such as commercial exchanges, its alignment might not be optimal for all systems' workflows that don't necessitate this level of transaction frequency (*Evermann & Kim, 2019*).

## Anonymity

Preserving the anonymity of blockchain transactions is fundamental to their acceptance within these frameworks. The challenge lies in striking a balance between maintaining privacy for both blockchain participants and transactions, while enabling verification by other members for blockchain updates. The existing pseudonymous nature of most blockchain-based e-cash protocols falls short of providing comprehensive identity and transaction privacy. While anonymity is a pivotal trait of blockchain technology, safeguarding users from revealing their actual identities, it also evokes concerns, particularly in the context of money laundering. Anonymity empowers individuals to conduct global fund transfers without traceable evidence beyond wallet addresses. Consequently, the blockchain has gained favor among cybercriminals engaged in money laundering endeavors (*Andola et al., 2021*).

## Immutability

Blockchain technology inherently embodies immutability, a property wherein recorded information becomes unalterable once committed to the blockchain. This property aligns logically with the design of systems. However, in the context of archives, which house records susceptible to long-term changes, the concept of immutable data presents a dual challenge. Archives necessitate the capacity to modify records' metadata, ensuring both authenticity after digital preservation actions and the preservation of relationships with subsequent records introduced after an initial record's entry into the archive, registered within a blockchain. The imperative to maintain archival bonds, signifying networks of relationships among aggregated records, exemplifies this requirement. Thus, the dichotomous nature of immutability mandates judicious contemplation, encompassing both its merits and limitations within archival frameworks (*Stančić & Bralić, 2021*; *Politou et al., 2019*; *Hughes et al., 2019*).

## Storage problems

In a blockchain system, like Bitcoin for instance, each node operates independently without needing a central authority. Every node stores a complete record of all transactions in a database. However, this decentralized setup leads to a notable outcome—the transaction database grows rapidly over time. As the system keeps working, the memory capacity of each node has to keep expanding to handle its operations smoothly. This becomes even more crucial in the context of today's huge data era, where more network activity results in transactions happening faster. This means that nodes that hold all the data (full nodes) need more memory to make sure transactions are checked properly. And in the era of big data, where there's a lot of information being exchanged, the number of nodes connected to the blockchain network is also increasing, leading to even more growth in the blockchain's transaction database. This poses challenges because the more users there are, the more data there is to store within the blockchain system (*Xu et al., 2020*; *Jia et al., 2021*; *Zhang et al., 2021*).

### Real-time monitoring and full provenance

A blockchain operates through a network of peers engaging in Peer-to-Peer (P2P) transactions. Peers record transactions within a given time span and bundle them into a block for incorporation into the blockchain. The decentralized nature of blockchain ensures resilience against tampering and facilitates traceability. Smart contracts, integral to blockchain, are coded agreements governed by event triggers. As no central entity governs them, they inherently engender trust (*Zheng et al., 2018a*). The invocation of a blockchain smart contract occurs by dispatching a "transaction"—termed the invoking transaction—to validating peers. This transaction comprises the contract's address, the calling function, and parameters. Upon receipt, peers execute the smart contract independently. Subsequently, consensus is achieved among distinct peers through consensus protocols, culminating in the recording of the execution outcome in the blockchain. Blockchain-based smart contracts offer enhanced trustworthiness, reduced reliance on centralized authorities, and expansive applicability. This decentralized framework functions without central authority. Nevertheless, instances arise necessitating real-time monitoring between regulators and participants, and an exhaustive record of past transactions and assets remains vital (*Helo & Shamsuzzoha, 2020*).

### Other considerations

When considering the use of blockchain, there are important things to think about. These factors include needing a unique digital identifier that should work worldwide, a decentralized naming service, a way to securely show who owns what, and making it easier to solve problems and disagreements without manual effort.

## CONCLUSIONS

In conclusion, blockchain technology holds tremendous potential to transform various industries by providing a secure, transparent, and decentralized system for data management and transaction processing. The inclusion of smart contracts has made blockchain technology even more intelligent, intricate, and automated. The integration of existing scientific research into the blockchain system is feasible and can take its application to the next level. Although the technology has already demonstrated its value in sectors such as finance, supply chain management, and healthcare, its utilization is expected to expand further in the future. Blockchain's unique features, such as distributed storage, decentralized management, shared maintenance, consensus trust, and a reliable database, can help overcome many of the challenges posed by conventional centralized systems, including security, transparency, and efficiency. However, there are still technical and regulatory challenges that need to be addressed before blockchain technology can be fully integrated into mainstream systems. Nevertheless, the growing adoption of blockchain by major corporations and governments globally confirms its potential as a disruptive technology that can reshape the way we store, manage, and exchange data. While blockchain technology is already being utilized in some exclusive domains, its application and development will undoubtedly require a considerable amount of time. As with other emerging technologies, it is vital to gather experience and knowledge to refine

and enhance blockchain's capabilities. Additionally, extensive research, experimentation, and innovation will be necessary to overcome potential limitations and challenges, ensuring its continued progress and success.

## ACKNOWLEDGEMENTS

The authors thank the anonymous reviewers for their comments and suggestions.

### Funding

This article is supported by the project supported by the Key Scientific Research Projects of Colleges and Universities in Henan Province (Grand No. 23A520054), and the Open Foundation of State Key Laboratory of Networking and Switching Technology (Beijing University of Posts and Telecommunications) (KLNST-2020-2-01). The funders had no role in study design, data collection and analysis, decision to publish, or preparation of the manuscript.

### Grant Disclosures

The following grant information was disclosed by the authors:
Key Scientific Research Projects of Colleges and Universities in Henan Province: 23A520054.
Networking and Switching Technology (Beijing University of Posts and Telecommunications): KLNST-2020-2-01.

### Competing Interests

Shi Dong is an Academic Editor for PeerJ.

### Author Contributions

- Shi Dong conceived and designed the experiments, prepared figures and/or tables, authored or reviewed drafts of the article, and approved the final draft.
- Khushnood Abbas conceived and designed the experiments, authored or reviewed drafts of the article, and approved the final draft.
- Meixi Li conceived and designed the experiments, prepared figures and/or tables, authored or reviewed drafts of the article, and approved the final draft.
- Joarder Kamruzzaman conceived and designed the experiments, authored or reviewed drafts of the article, and approved the final draft.

### Data Availability

  This is a literature review.

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
