# Peer review of "Blockchain technology and application: an overview"

_PeerJ Computer Science, doi:10.7717/peerj-cs.1705_

## Round 0.1 · original submission · Major Revisions

All the reviewers have raised valid concerns and need to be answered. Particularly, more in-depth analysis of block chain technology should be carried out and, its limitations and applicability should be discussed.

Reviewer 1 ·

Basic reporting

The structure of the article can be enhanced by implementing a more cohesive framework. For example, introducing an additional layer of categorization in the section titled "Key Technologies for Blockchain" would enhance clarity and logical progression.

There is inconsistency in the capitalization of headers, with some headers in all caps, followed by a mix of capitalizing only the first letters of each word, and then returning to all caps for the final headers.

There is inconsistency in the use of italic font and tabs at many places.

The figures are not appropriately placed; for instance, Figure 4, which should have been under 'Blockchain Wallet,' is instead placed under 'Proof of Work’. The adjustment will enhance the clarity.

Experimental design

The content within "The Advantages and Disadvantages of Blockchain Technology" section primarily focuses on different types of attacks. It would be more appropriate to change the header to a title that reflects this emphasis on attacks rather than general advantages and disadvantages.

After line 165, a formula q = kp is provided. However, the meaning of p is not explained in the surrounding context

Please cite the relevant source to mention the Consortium or Federated Blockchain. Additionally, it is important to note that consortium blockchains are typically considered semi-centralized rather than fully decentralized - line 331

Validity of the findings

The repeated mention of China in the introduction gives the impression that the rest of the paper will primarily focus on China. It would be beneficial to consolidate the three sentences and make them more generic to avoid creating a specific geographical emphasis.

Additional comments

I commend the authors for the comprehensive study of blockchain, covering various aspects from advantages to challenges. However, I have noticed a lack of novelty in the article, particularly in the "Introduction to Future Research Methods and Application Areas" section. It would be beneficial to consolidate the two topics on IoT into one, allowing for a more comprehensive exploration. Similarly, the section on AI also missed the opportunity to explore the potential value of decentralization. It would have been interesting to see how decentralization could enhance AI.

Cite this review as
Anonymous Reviewer (2023) Peer Review #1 of "Blockchain technology and application: an overview (v0.1)". PeerJ Computer Science

Reviewer 2 ·

Basic reporting

• The manuscript is well written and deserves publication in my opinion. However, the following changes will enhance the overall quality further.
• Authors must divide the introduction into separate paragraphs highlighting the background, motivation, major contribution and paper organization.
• Present a comparison of various algorithms in section 2. Refer to Blockchain based solutions to Secure Iot: Background, integration trends and a way forward.
• Summarize the various types of attacks discussed in advantage and disadvantage of blockchain section.
• Highlight the role of cryptography in blockchain technology. Refer to Unification of Blockchain and Internet of Things (BIoT): Requirements, working model, challenges and future directions.
• Provide sufficient evidence for “The primary manifestation of the security of blockchain technology is its ability to protect information interactions effectively against human intervention”.
• Limitations and applicability of blockchain needs to be discussed in more detail.

Experimental design

• Summarize the various types of attacks discussed in advantage and disadvantage of blockchain section.
• Highlight the role of cryptography in blockchain technology. Refer to Unification of Blockchain and Internet of Things (BIoT): Requirements, working model, challenges and future directions.
• Provide sufficient evidence for “The primary manifestation of the security of blockchain technology is its ability to protect information interactions effectively against human intervention”.
• Limitations and applicability of blockchain needs to be discussed in more detail.

Validity of the findings

NA

Additional comments

• The manuscript is well written and deserves publication in my opinion. However, the following changes will enhance the overall quality further.
• Authors must divide the introduction into separate paragraphs highlighting the background, motivation, major contribution and paper organization.
• Present a comparison of various algorithms in section 2. Refer to Blockchain based solutions to Secure Iot: Background, integration trends and a way forward.
• Summarize the various types of attacks discussed in advantage and disadvantage of blockchain section.
• Highlight the role of cryptography in blockchain technology. Refer to Unification of Blockchain and Internet of Things (BIoT): Requirements, working model, challenges and future directions.
• Provide sufficient evidence for “The primary manifestation of the security of blockchain technology is its ability to protect information interactions effectively against human intervention”.
• Limitations and applicability of blockchain needs to be discussed in more detail.

Cite this review as
Anonymous Reviewer (2023) Peer Review #2 of "Blockchain technology and application: an overview (v0.1)". PeerJ Computer Science

Reviewer 3 ·

Basic reporting

The paper is written in professional English and is easy to read throughout.

Some of the sections of the paper are missing references. For example, statements such as "While blockchain technology is still in its developmental stage, a blockchain industry has been formed." are not matching the context of the corresponding section.

Figure/table quality should be improved. Figure 5 is not readable.

There are review papers on blockchain, and this paper reads more like a tutorial. Hence, additional enhancements are necessary to determine the full impact of the paper and improve its research aspects.

The introduction needs to be improved. Currently, it provides a narrow view of blockchain technology. Hence, the motivation is not clear.

Experimental design

While China may be a leader in blockchain technology, authors should provide global examples to increase awareness and motivation. Besides, references must be provided for the corresponding examples/statements.

"Key Technologies for Blockchain" section is missing many references. Besides, the explanations/descriptions are too generic. A person knowledgeable in blockchain may understand the context. However, the authors fail to explain why these details are necessary, causing confusion for those unfamiliar with the domain. The same issues applies to the "Blockchain applications" section.

The paper's research aspect requires improvement. Currently, it only provides a basic overview of the field.

It would be appropriate to change the title of the section "Social Life" to a more fitting one.

The authors should provide a proper explanation for the term "the traceability industry". It may be helpful to phrase it appropriately.

Validity of the findings

The content must be more specific. For instance, under "Phishing" (line 615), it only provides a general explanation. It should explain how the attack affects blockchain. This is noticeable in other sections as well. Besides, some sections, like "Dictionary attacks," are not unique to blockchain and can provide redundant information about the domain.

Many parts of the article seem too broad and lack specificity, resembling a magazine paper.

Cite this review as
Anonymous Reviewer (2023) Peer Review #3 of "Blockchain technology and application: an overview (v0.1)". PeerJ Computer Science

---

## Round 0.2 · Minor Revisions

Please read the reviewers' comments carefully and revise the paper accordingly. Particular, improve the Sections 1, and 2 and the quality of figures needs improvements. Comparison with the existing works should also be considered.

Reviewer 2 ·

Basic reporting

The article is revised and can be accepted for publication after some minor changes.
1. Section 1 needs to highlight the background and problem statement properly.
2. Section 2 still looks weak and must be significantly ameliorated.
3. Comparison with existing works in needed.

Experimental design

no comment

Validity of the findings

no comment

Additional comments

no comment

Cite this review as
Anonymous Reviewer (2023) Peer Review #2 of "Blockchain technology and application: an overview (v0.2)". PeerJ Computer Science

Reviewer 3 ·

Basic reporting

The figure quality is still poor. Figure 5 needs further enhancements (e.g., try to increase the font size).

The references are still inadequate and require updating.

Experimental design

The authors have improved the paper based on the comments.

Validity of the findings

The authors have improved the paper according to the comments.

Additional comments

No

Cite this review as
Anonymous Reviewer (2023) Peer Review #3 of "Blockchain technology and application: an overview (v0.2)". PeerJ Computer Science

---

## Round 0.3 · Minor Revisions

The quality of the figures still needs improvement. Sections 1 and 2 need to be improved as per the reviewers' earlier suggestions

---

## Round 0.4 · accepted · Accept

The authors have now addressed all the comments raised by the reviewers.